# Structure-based insights into self-cleavage by a four-way junctional twister-sister ribozyme

Luqian Zheng[1], Elisabeth Mairhofer[2], Marianna Teplova[3], Ye Zhang[1], Jinbiao Ma [4,5], Dinshaw J. Patel[3], Ronald Micura[2] & Aiming Ren[1,5]

Here we report on the crystal structure and cleavage assays of a four-way junctional twister-sister self-cleaving ribozyme. Notably, 11 conserved spatially separated loop nucleotides are brought into close proximity at the ribozyme core through long-range interactions mediated by hydrated $Mg^{2+}$ cations. The C62–A63 step at the cleavage site adopts a splayed-apart orientation, with flexible C62 directed outwards, whereas A63 is directed inwards and anchored by stacking and hydrogen-bonding interactions. Structure-guided studies of key base, sugar, and phosphate mutations in the twister-sister ribozyme, suggest contributions to the cleavage chemistry from interactions between a guanine at the active site and the non-bridging oxygen of the scissile phosphate, a feature found previously also for the related twister ribozyme. Our four-way junctional pre-catalytic structure differs significantly in the alignment at the cleavage step (splayed-apart vs. base-stacked) and surrounding residues and hydrated $Mg^{2+}$ ions relative to a reported three-way junctional pre-catalytic structure of the twister-sister ribozyme.

[1] Life Sciences Institute, Zhejiang University, 310058 Hangzhou, China. [2] Institute of Organic Chemistry, Leopold Franzens University, A6020 Innsbruck, Austria. [3] Structural Biology Program, Memorial Sloan-Kettering Cancer Center, New York, NY 10065, USA. [4] Department of Biochemistry, State Key Laboratory of Genetic Engineering, Institute of Plant Biology, School of Life Sciences, Fudan University, 200438 Shanghai, China. [5] Collaborative Innovation Centre of Genetics and Development, Fudan University, 200438 Shanghai, China. Luqian Zheng and Elisabeth Mairhofer contributed equally to this work. Correspondence and requests for materials should be addressed to R.M. (email: ronald.micura@uibk.ac.at) or to A.R. (email: AimingRen@zju.edu.cn)

There has been considerable interest towards an improved understanding of the cleavage mechanisms that underlie RNA catalysis by small self-cleaving ribozymes involved in self-scission during rolling circle replication of viral genomes[1, 2]. Such challenges are highlighted by the recent identification of a new set of self-cleaving ribozymes from genomic bioinformatics searches, named twister, twister-sister, pistol, and hatchet ribozymes[3, 4]. These nucleolytic ribozymes site-specifically cleave phosphodiester linkages by activating the 2′-OH of the ribose 5′- to the scissile phosphate for inline attack on the adjacent P-O5′ bond to yield 2′,3′-cyclic phosphate and 5′-OH ends. The key mechanistic issues relate to the geometry of the dinucleotide step including inline alignment at the cleavage site, as well as the contributions of nearby nucleobases, ribose 2′-OH and hydrated $Mg^{2+}$ cations in deprotonating the 2′-OH nucleophile, stabilizing the transition state, and protonating the oxyanion leaving group[5]. In addition, clarification is needed regarding the nature and extent of commonalities/differences in

RNA catalysis mechanisms between small self-cleaving ribozymes[6, 7].

The earliest studies on recently identified small self-cleaving ribozymes focused on the twister ribozyme and included structural[8–10], mechanistic[11, 12], and computational[13, 14] contributions. The currently available structures of twister ribozymes reflect pre-catalytic conformations that display distinctions at the cleavage site with respect to the attacking 2′-O P-O5′ alignment and the presence/absence of hydrated $Mg^{2+}$ cations at the scissile phosphate. All structures are in agreement with the conclusion that the pre-catalytic fold of the twister ribozyme provides conformational flexibility of the pyrimidine nucleoside at the pyrimidine–purine cleavage step in contrast to the anchored purine nucleoside, suggestive of a likely realignment of the pyrimidine and its 2′-OH to achieve inline alignment in the transition state. Interestingly, this ribozyme can cleave a single nucleoside (5′ of the scissile phosphate) with only twofold reduced rate and despite the inability to form the phylogenetically

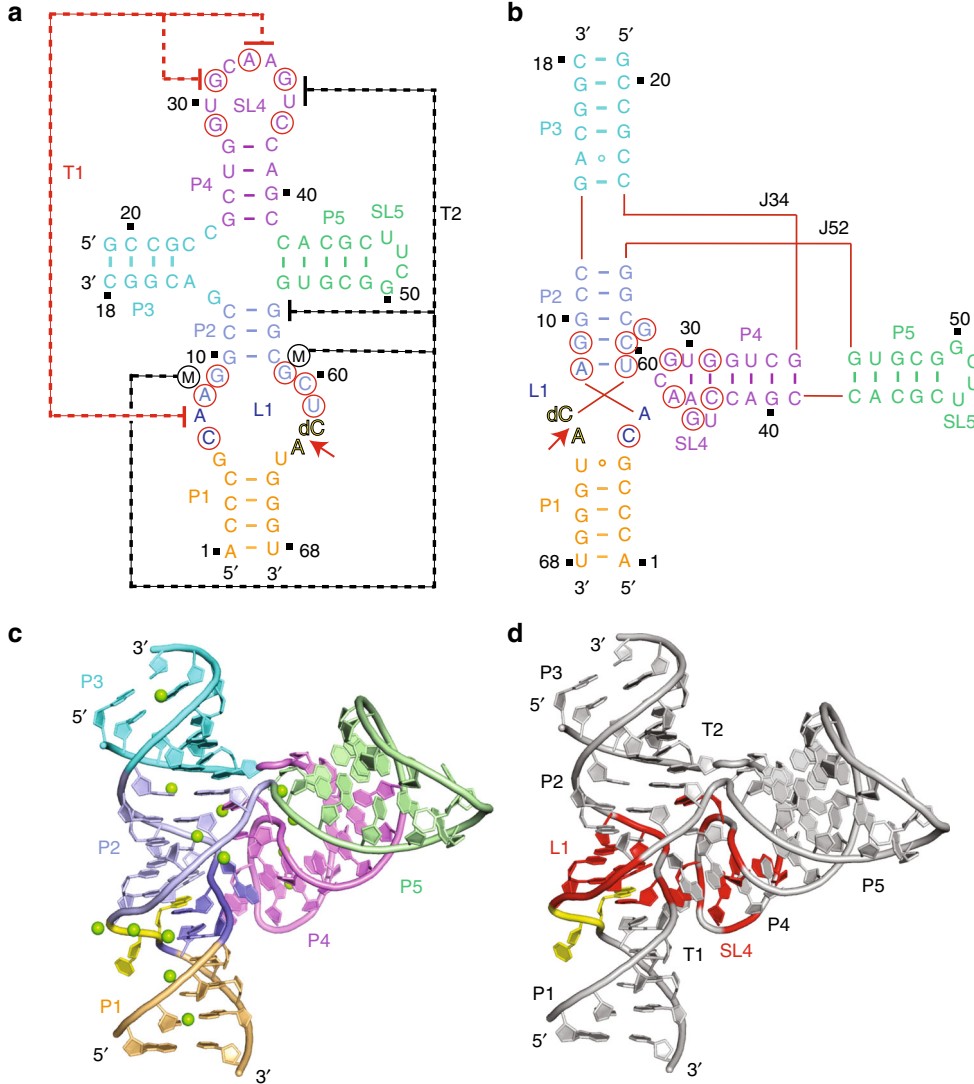

**Fig. 1** Schematic and tertiary structure of the twister-sister ribozyme. **a** Schematic of the secondary fold of the dC62-containing four-way junctional twister-sister ribozyme. The sequence is color-coded according to helical segments observed in the tertiary structure. All highly conserved residues are labeled by red circles. Long-distance interactions observed in the tertiary structure are labeled with dashed lines named T1 (in red) and T2 (in black). The cleavage site step was shown by a red arrow. **b** Schematic of the tertiary fold based on the crystal structure of the dC62-containing four-way junctional twister-sister ribozyme. **c** A ribbon view of the 2 Å structure of the four-way junctional twister-sister ribozyme color-coded as shown in **a** and **b**. The divalent metal ions identified in the tertiary structure are shown as green balls. **d** Highly conserved residues (shown in red) are brought into close proximity by the interaction between partially zipped-up L1 and SL4 loops in the tertiary fold of the twister-sister ribozyme

conserved stem P1[11]. A single-molecule FRET study on twister folding rationalizes this behavior by revealing that the active site-embracing pseudoknot fold is preserved in the shortened ribozyme and also in the cleaved 3′-product RNA that both lack P1[12]. More recently, the structural studies have been extended to the pistol ribozyme[15, 16], where there is good consensus on both the overall folding topology and alignment of catalytic residues at the cleavage site. This may reflect the anchoring of both pyrimidine and purine at the pyrimidine–purine cleavage step in the precatalytic conformation of the pistol ribozyme.

Twister and twister-sister appear to be related ribozymes based on similar helical connectivities and common conserved residues positioned in similarly aligned hairpin and internal loops[4]. We report here the structure of a four-way junctional twister-sister ribozyme and discuss possible/putative roles of selected nucleobases and hydrated $Mg^{2+}$ ions observed at the active site for cleavage chemistry. We also compare our structure of twister-sister ribozyme composed of a four-way junctional fold (this study), with a recently reported structure of the twister-sister ribozyme composed of a three-way junctional fold[17] and note unanticipated differences in one long-range interaction and in alignments in the catalytic pocket between the two structures. Finally, a comparison between previously reported structures of the twister ribozyme[9, 11] and the structure of the twister-sister ribozyme (this study) highlights similarities in structural alignments in the catalytic pocket that may indicate shared concepts of these two related ribozymes to mediate cleavage by involving a guanine–scissile phosphate interaction, a splayed-apart conformation of the cleavage site, and involvement of—although to different extent—hydrated divalent $Mg^{2+}$ ions.

## Results

**Twister-sister ribozyme construct and cleavage propensity**. We screened a large number of chemically synthesized two-stranded and three-stranded constructs of the twister-sister ribozyme, before identifying a two-stranded construct composed of a four-way junctional fold shown schematically in Fig. 1a, that yielded diffraction quality crystals. Cleavage assays on the wild-type two-stranded twister-ribozyme construct established site-specific cleavage at the C62–A63 step under 2 mM divalent cation conditions with fastest cleavage observed for $Mn^{2+}$, followed by $Mg^{2+}$ and $Ca^{2+}$; cleavage was slowest for $Sr^{2+}$ (Supplementary Fig. 1a, b).

**Tertiary fold of the twister-sister ribozyme**. For crystallographic studies, we introduced dC62 into the sequence of the longer substrate strand of the two-stranded construct, so as to prevent cleavage at the known C62–A63 cleavage site of the twister-sister ribozyme[4]. The structure was solved at 2 Å resolution using phases determined for crystals soaked with $Ir(NH_3)_6^{3+}$ (X-ray statistics are listed in Supplementary Table 1). There are two molecules of the twister-sister ribozyme in each asymmetric unit (space group: $P2_12_12$), with both exhibiting well-defined electron density, while also superpositioning well with an rmsd = 0.27 Å. A schematic of the tertiary fold of the twister-sister ribozyme is shown in Fig. 1b, whereas the three-dimensional structure in a ribbon representation is shown in Fig. 1c.

As shown in the schematic of the secondary fold in Fig. 1a, our two-stranded construct of the twister-sister ribozyme is composed of stems P1 (in orange), P2 (in blue) and P3 (in cyan), and stem-loops P4-SL4 (in magenta) and P5-SL5 (in green). Loop L1 is positioned between stems P1 and P2, whereas a four-stem junction is formed at the intersection of stems P2, P3, P4, and P5. Highly conserved residues are shown in red circles and are distributed between L1 and SL4 loop segments, which are spatially separated in the secondary structure (Fig. 1a) but

brought into close proximity in the tertiary structure (Fig. 1d; residues labeled in red) of the twister-sister ribozyme.

As shown in the schematic of the tertiary fold of our construct of the twister-sister ribozyme (Fig. 1b), stems P1, P2, and P3 form one continuous helix (Supplementary Fig. 2a), whereas stems P4 and P5 form a second continuous helix (Supplementary Fig. 2b), with elements of these two continuous helices meeting at the junctional site (Fig. 1b, d). Notably, SL4 projecting from continuous helix P5–P4 is positioned in the minor groove of the partially zippered-up L1 loop segment associated with the continuous helix P1–P2–P3 helix (Fig. 1b, c). The stacking between terminal base pairs of extended stems P1 and P2 are shown in Supplementary Fig. 2c and d, those between extended stems P2 and P3 are shown in Supplementary Fig. 2e and f, whereas those between terminal base pairs of stems P4 and P5 are shown in Supplementary Fig. 2g and h.

We observe additional pairing on formation of the tertiary fold of the twister-sister ribozyme (Fig. 1b). Thus, loop L1 partially zippers up, whereby stem P1 is extended through *trans* non-canonical G5•C64 (Supplementary Fig. 3a) and *trans* sugar edge-Hoogsteen C6•A63 (Supplementary Fig. 3b) pairing involving G5 and C6, whereas stem P2 is extended by the Watson–Crick A8•U61, which forms part of a major groove aligned G59•(A8–U61) base triple (Supplementary Fig. 3c) and the Watson–Crick G9•C60 pairing, accompanied by extrusion of G59. Note that all residues of the base triple, namely A8, G59, and U61 are highly conserved in the twister-sister ribozyme. SL4 also partially zippers up, whereby stem P4 is extended through the Watson–Crick G29•C37 pairing involving highly conserved residues and the *trans* Watson–Crick–Hoogsteen U30•A34 pairing (Supplementary Fig. 3d), accompanied by extrusion of G35 and U36. In addition, stem P3 is extended by the *cis* Watson–Crick–Watson–Crick A14•C23 (Supplementary Fig. 3e) and Watson–Crick G13•C24 pairing at the four-way junctional site of the twister-sister ribozyme.

We have identified a large number of hydrated $Mg^{2+}$ cations distributed throughout the tertiary fold of the twister-sister ribozyme (green balls, Fig. 1c), with divalent cation $Mg^{2+}$ occupancy confirmed followed substitution by $Mn^{2+}$ and monitoring the latter's anomalous diffraction characteristics (Supplementary Fig. 4a). Our studies have focused on seven $Mg^{2+}$ cation sites, all of which have been validated by $Mn^{2+}$ replacement, as shown for divalent cations labeled M1 to M4 in Supplementary Fig. 4b and c, for M5 in Supplementary Fig. 4d and e, and for M6 to M7 in Supplementary Fig. 4f and g.

**Structural alignments at the four-way junctional site**. At the junctional site of the twister-sister ribozyme, both splayed-apart C24–G25 (junctional J34 segment) and G55–G56 (junctional J52 segment) steps (Fig. 1b) are brought into close proximity, with a hydrated $Mg^{2+}$ (labeled M5) bridging opposingly aligned J34 and J52 junctional segments (Fig. 2a). A network of hydrogen bonds involving C24–G25, G55–G56, looped out highly conserved G35 and hydrated M5 stabilizes this junction site (Fig. 2a, b). M5 forms symmetrical inner-sphere coordination to the non-bridging *pro-S*$_P$ oxygen of the C24–G25 phosphate and the *pro-S*$_P$ oxygen of the G55–G56 phosphate, and to four water molecules (Fig. 2b). Notably, of the five nucleotides (C24, G25, G55, G56, and G35) involved in the alignment at this junctional site, only C24 adopts a C2′-*endo* sugar pucker conformation. C24 and G56 form terminal stacking interactions between stems P2 and P3 (Fig. 2a; Supplementary Fig. 2e, f), whereas G25 and G55 form terminal stacking interactions between stems P4 and P5 (Fig. 2a; Supplementary Fig. 2g, h).

The A33–A34 step of stem-loop SL4 (composed of loop G31–C32–A33 segment closed by a non-canonical U30•A34

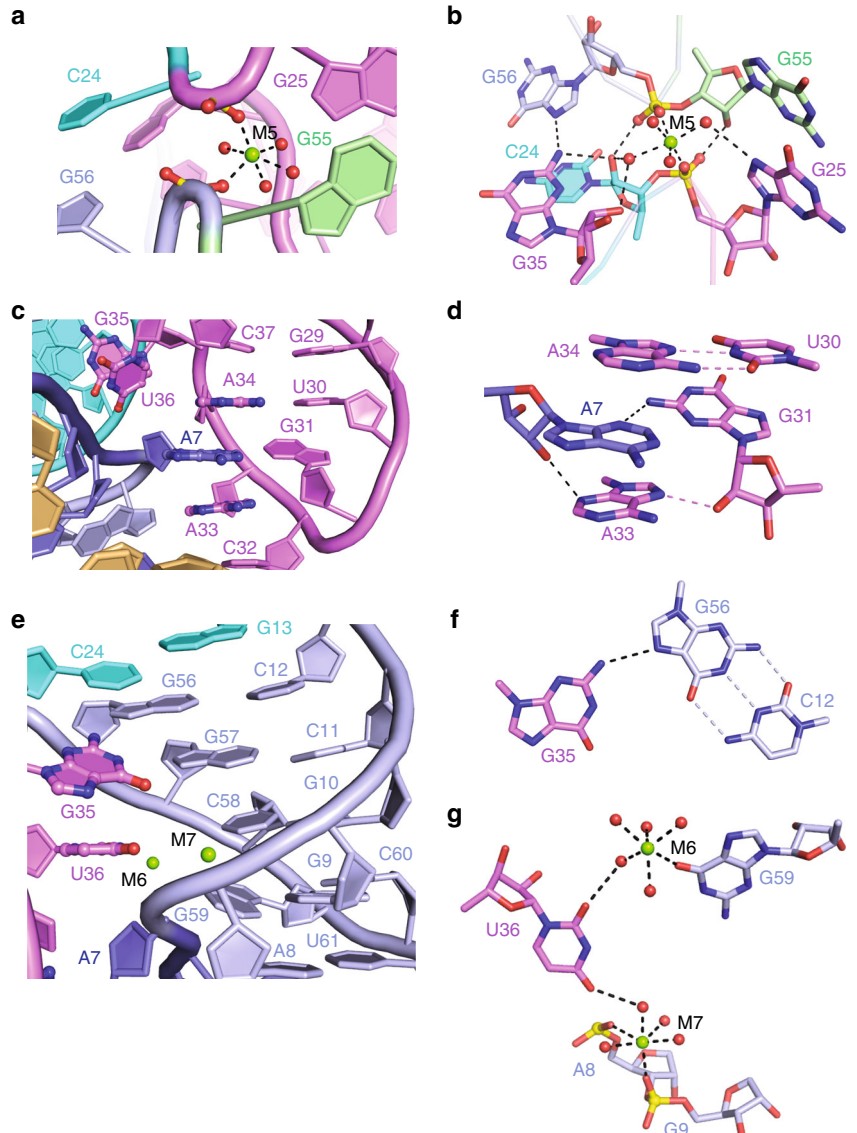

**Fig. 2** Tertiary interactions in the structure of the twister-sister ribozyme. **a**, **b** Tertiary interactions formed at the four-way junction. One hydrated $Mg^{2+}$ (labeled M5) is coordinated directly to the *pro-S*$_P$ non-bridging phosphate oxygens of G55–G56 and C24–G25 steps, whereas the *pro-R*$_P$ non-bridging phosphate oxygens form hydrogen bonds with the 2′-OH groups of G55 and C24. One inner-sphere hydration water of M5 forms a hydrogen bond with N7 of G25, whereas another inner-sphere hydration water of M5 forms two hydrogen bonds with G35. **c**, **d** Continuous stacking interactions within the four-way junctional region. A7, which is extruded from L1, forms stacking (**c**) and hydrogen bond (**d**) interactions with the zipped-up SL4 loop segment. **e–g** Stacked bases G35 and U36, which are extruded from SL4, interact with stem P2 and L1 (**e**). Extruded G35 forms one hydrogen bond with the major groove edge of the Watson–Crick G56–C12 pair (**f**). Extruded U36 forms two hydrogen bonds with the inner-sphere waters of two hydrated $Mg^{2+}$ ions, in which one is coordinated with O6 of G59 and the other is coordinated with the non-bridging phosphate oxygens of residues A8 and A9 (**g**)

pair) forms a long-range interaction with extruded A7 from the zipped-up loop L1 (see red dashed line labeled T1 in Fig. 1a), whereby A7, which adopts a C2′-*endo* sugar pucker conformation, is sandwiched between adjacent A33 and A34 (Fig. 2c), and anchored in place by forming a non-canonical A7•G31 pair (Fig. 2d). Notably, the G31–C32–A33 segment and A7 adopts a stable GNRA hairpin loop fold. The major groove base edge of highly conserved residue A33 forms a hydrogen bond with the 2′-OH of G31, whereas its minor groove base edge forms a hydrogen bond with the 2′-OH of A7, contributing to the hydrogen bond network stabilizing the interaction between A7, G31, and A33 (Fig. 2d).

Further, looped out bases G35 and U36 emanating from zipped-up SL4 loop adopt a stacked alignment and are positioned in the major groove of stacked bases of stem P2 and

zipped-up loop L1, thereby forming a second long-range alignment (Fig. 2e; see black dashed line labeled T2 in Fig. 1a), with highly conserved G35 anchored in place through formation of a major groove aligned G35•(G56–C12) base triple (Fig. 2f). In addition, we found that the carbonyl groups of U36 forms two hydrogen bonds with the inner-sphere water of two separate hydrated $Mg^{2+}$ ions labeled as M6 and M7 in Fig. 2g. M6 is directly coordinated with O6 of G59 and M7 is directly coordinated with the non-bridging phosphate oxygens of A8 and G9. Notably, both U36 and G59 adopt C2′-*endo* sugar pucker conformations in the structure. The involved residues G59, A8, and G9 and the surrounding interacting residues C6, C60, and U61 are all highly conserved in the sequence of the twister-sister ribozyme (Figs. 1a and 2g; Supplementary Fig. 2c, d), which is consistent with the importance of the long-distance

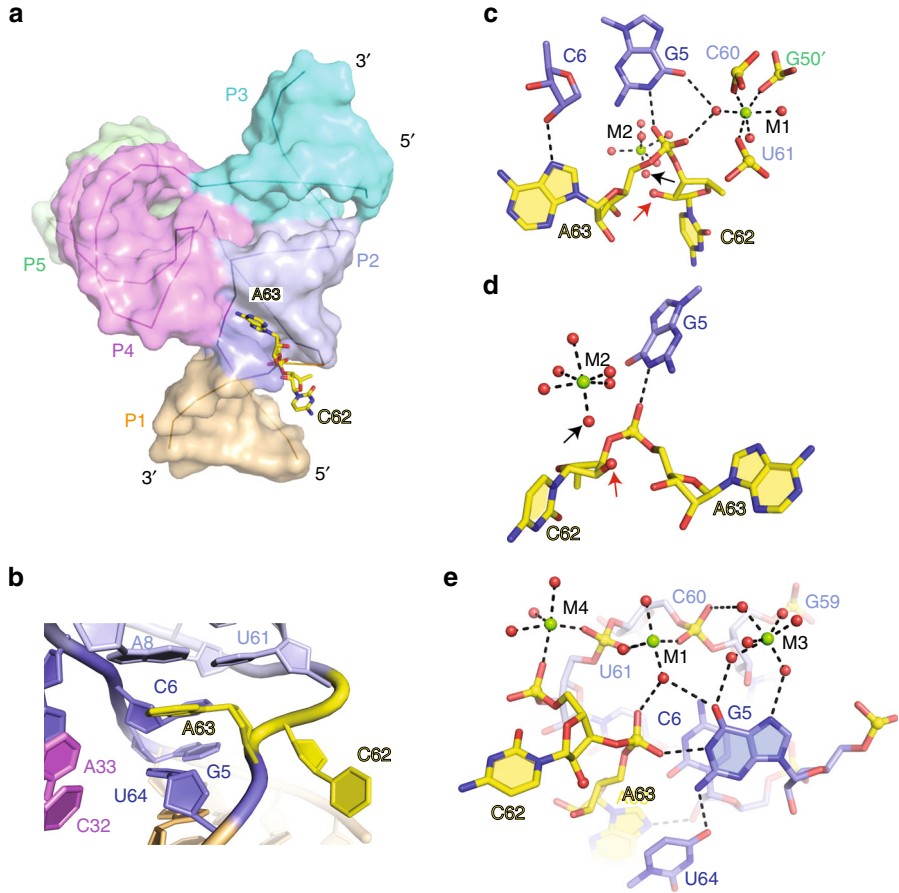

**Fig. 3** Intermolecular contacts and bound $Mg^{2+}$ ions at the C62–A63 cleavage site in the structure of the twister-sister ribozyme. **a** The structure of the four-way junctional twister-sister ribozyme with the RNA in a surface representation, except for the splayed apart C62–A63 site, which is shown in a stick representation. **b** Bases C62 and A63 at the cleavage site are splayed apart, with C62 directed outwards and A63 directed inwards into the ribozyme fold. A63 (*anti* alignment) is anchored in place by stacking between Watson–Crick A8–U61 and non-canonical *trans* U64•G5 pairs. **c** The non-bridging phosphate oxygens of the C62–A63 scissile phosphate form hydrogen bonds to the N1H of G5 and an inner-sphere water of hydrated $Mg^{2+}$ labeled M1. In addition, hydrated $Mg^{2+}$-labeled M1 is coordinated to *pro-R*$_P$ oxygen of U61 (2.1 Å), to the *pro-R*$_P$ of C60 (2.2 Å) and a site (G50′) on a symmetry-related molecule in the crystal lattice. A hydrated $Mg^{2+}$-labeled M2 was identified close to the modeled 2′-OH of C62. Further, N7 of A63 forms a hydrogen bond with the 2′-OH of C6. A red arrow indicates the modeled 2′-O of C62, whereas a black arrow labels the inner-sphere water of M2 closest to the modeled 2′-O of C62. **d** An alternate and simplified view of the drawing in **c**. **e** The octahedral coordination geometries and hydrogen bond networks involving a set of three divalent $Mg^{2+}$ ions labeled as metals M1, M3, and M4 in the vicinity of the cleavage site within a single twister-sister ribozyme. Both $Mg^{2+}$ ions labeled M1 and M4 are also coordinated to sites on a symmetry-related molecule in the crystal lattice (not shown)

interaction T2, and its potential to impact on formation of the active site.

**Structural alignments at the C62–A63 cleavage step**. Bases C62 and A63 at the cleavage site are splayed apart, with C62 directed outwards and A63 directed inwards into the ribozyme fold (Fig. 3a). C62 appears to adopt a mixture of *syn* and *anti* conformations reflective of flexibility about its glycosidic bond, whereas A63 is anchored in place in an *anti* conformation by stacking between bases (Fig. 3b), with its N7 atom hydrogen bonded to the 2′-OH of C6 (Fig. 3c). We have modeled the oxygen corresponding to 2′-OH at position C62 and measure a distance of 3.8 Å between the 2′-O and the phosphorus of the C62–A63 cleavage step and an angle of 71° for 2′-O of C62 relative to the P-O5′ bond (off-line alignment) at the cleavage step.

The non-bridging oxygens of the C62–A63 scissile phosphate are anchored in place through hydrogen bonding to the N1H of G5 (coordination to *pro-S*$_P$ oxygen with heteroatom separation of 2.7 Å) and to an inner-sphere water of a hydrated $Mg^{2+}$ labeled M1 (H-bonded to *pro-R*$_P$ oxygen with heteroatom separation of

2.9 Å; M1 is also coordinated to *pro-R*$_P$ oxygen of U61 (2.1 Å) and *pro-R*$_P$ of C60 (2.2 Å); it is further coordinated to *pro-Sp* of G50′ from a symmetrical molecule (2.3 Å)) (Fig. 3c). The modeled 2′-OH of C62 (red arrow, Fig. 3c, d) is positioned for hydrogen bonding to an inner-sphere water of hydrated $Mg^{2+}$-labeled M2 (heteroatom separation of 3 Å) (black arrow, Fig. 3c, d).

The octahedral coordination geometries and hydrogen bond networks of a set of three hydrated $Mg^{2+}$ cations labeled M3, M1, and M4 within a single twister-sister ribozyme are highlighted in Fig. 3e. Both M1 and M4 are also coordinated to groups in a symmetry-related molecule in the crystal lattice (coordination not shown in Fig. 3e) so as to maintain octahedral geometries for the divalent metal-ion sites.

**Cleavage assays on twister-sister ribozyme mutants**. We have undertaken cleavage assays in the context of a two-stranded construct of the twister-sister ribozyme (Fig. 4a; Supplementary Fig. 1a). Cleavage data on the wild-type twister-sister ribozyme under 2 mM $Mg^{2+}$, $Mn^{2+}$, $Ca^{2+}$, and $Sr^{2+}$ divalent cation conditions are compared in Supplementary Fig. 1b, with cleavage enhanced under $Mn^{2+}$ conditions relative to the other divalent

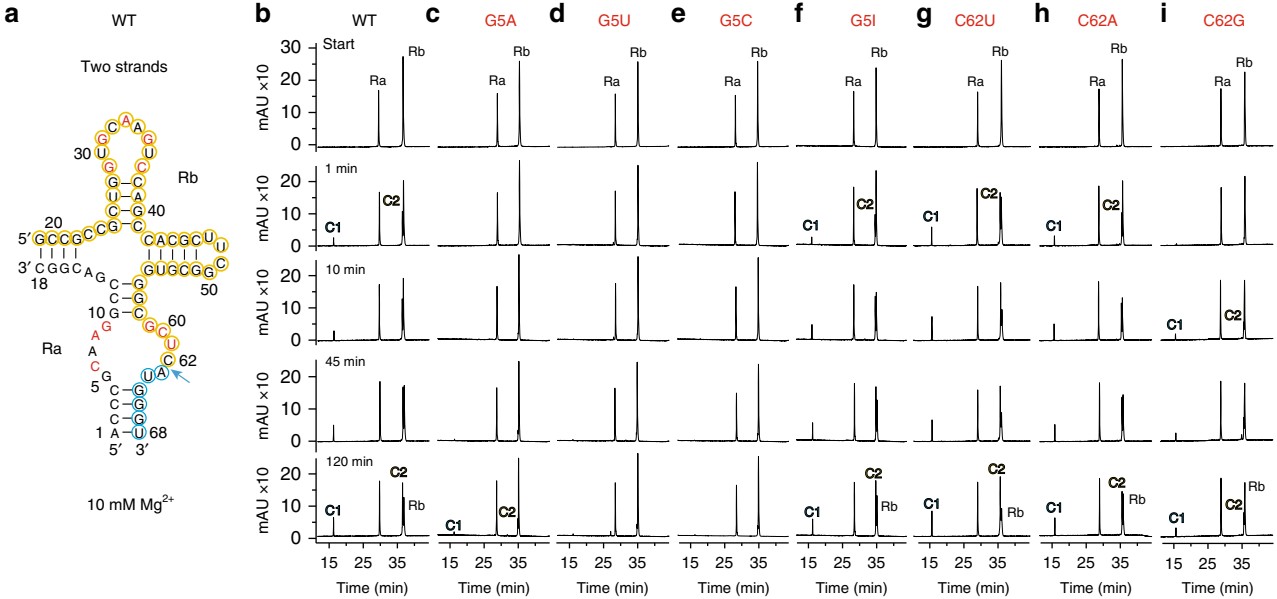

**Fig. 4** Self-cleavage of the two-stranded twister-sister ribozyme. **a** The secondary structure of the two-stranded twister-sister ribozyme used in the cleavage assays. Ra and Rb denote the 18-nt and 50-nt ribozyme strands; C1 (blue) and C2 (yellow) denote 6-nt and 44-nt cleavage products. **b–i** HPLC traces following cleavage activity of wild-type ribozyme (**b**) and mutants G5A (**c**), G5U (**d**), G5C (**e**), G5I (**f**), C62U (**g**), C62A (**h**), and C62G (**i**). Cleavage activity analyzed at 55 μM RNA each strand; 10 mM MgCl$_2$, 100 mM KCl, 30 mM HEPES, pH 7.5, 23 °C. HPLC conditions: Dionex DNAPac column (4 × 250 mm$^2$), 80 °C, 1 mL min$^{-1}$, 0–60% buffer B in 45 min. Buffer A: Tris–HCl (25 mM), urea (6 M), pH 8. Buffer B: Tris–HCl (25 mM), urea (6 M), NaClO$_4$ (0.5 M), pH 8

cations. In addition, we have also monitored cleavage data of G5 substitutions of the twister-sister ribozyme under 10 mM Mg$^{2+}$ conditions, with the data on G5A, G5U, G5C, and G5I shown in Fig. 4c–f. Although cleavage is essentially retained for the G5I substitution (Fig. 4f), there is essentially complete loss of cleavage for G5U and G5C substitutions (Fig. 4d, e) and very minor cleavage for G5A (Fig. 4c). We also performed the cleavage assays of the substitutions C62U (Fig. 4g), C62A (Fig. 4h), and C62G (Fig. 4i), with the first two mutants displaying pronounced cleavage activity, whereas the third mutant shows somewhat reduced cleavage activity.

Further comprehensive cleavage assays on base, sugar, and backbone modified twister-sister ribozymes have been undertaken on a three-stranded construct shown in Fig. 5a and Supplementary Fig. 1c, with the shorter individual strands facilitating site-specific incorporation of modifications. Cleavage assays on the three-stranded construct under 2 mM divalent cation conditions indicate no cleavage under Mg$^{2+}$ conditions (Supplementary Fig. 1d), whereas the onset of cleavage was observed under Mn$^{2+}$ conditions (Supplementary Fig. 1d). Importantly, cleavage of the three-stranded construct can also be observed under higher (10 and 20 mM) Mg$^{2+}$ conditions (Fig. 5b; Supplementary Fig. 1d).

Given these observations, we have undertaken cleavage assays on mutations of the three-stranded construct of the twister-ribozyme under 10 mM Mg$^{2+}$ conditions. These cleavage data are shown for wild-type (Fig. 5b) and substitutions G5A (Fig. 5c), C62A (Fig. 5d), C62G (Fig. 5e), A63U (Fig. 5f), A7U (Fig. 5g), A8U (Fig. 5h), as well as sugar modifications C6dC (Fig. 5i) and C24dC/G55dG (Fig. 5j) and backbone modification C60dCmP (Fig. 5k). For several substitutions/modifications, there is either partial (A8U and C6dC) (Fig. 5h, i) or complete loss (G5A, A63U, and C24dC/G55dG) (Fig. 5c, f, j) in activity. By contrast, we observe enhanced cleavage for C62A and C62G (Fig. 5d, e), cleavage comparable to wild-type for A7U (Fig. 5g), and a slight loss of cleavage extent for the C60dCmP substitution (Fig. 5k). Notably, although in the three-strand assay the C62A and C62G

mutants are of faster rate and higher yield compared with the wild-type (Fig. 5b, d, e), in the two-strand assay C62A shows very similar-to-wild-type behavior but C62G is slower in cleavage (Fig. 4b, h, i).

## Discussion

We first discuss our four-way junctional structure of the twister-sister ribozyme in the context of mutational studies and then compare and note unanticipated differences in the active site architecture with a recently solved three-way junctional structure of the twister-sister ribozyme[17]. We next compare our structures of the twister[9, 11] and twister-sister (this study) ribozymes to consider the merits of their designations as related ribozymes[4].

We have observed a large number of binding sites for hydrated metal cations in the structure of the twister-sister ribozyme (Fig. 1c), with the focus in this paper on metal sites labeled M1 to M7, positioned either at the junctional or cleavage sites. We have undertaken Mn$^{2+}$ soak experiments to use the anomalous properties of these ions to definitively differentiate between monovalent and divalent cation sites. These experiments (examples shown for M1 to M7 in Supplementary Fig. 4b–g) unambiguously establish that M1 to M7 are all divalent cations, with the observed octahedral geometry for M1 to M7 with metal to water oxygen distances of 2 Å, consistent with hydrated Mg$^{2+}$ for these metal sites (M1–M4 are shown in Fig. 3c–e; M5 is shown in Fig. 2a and b, and M6–M7 are shown in Fig. 2e and g).

One notable feature of the junctional alignment is the network of hydrogen-bonding interactions involving hydrated Mg$^{2+}$ cation M5-mediated stitching together of elements from stems P2, P3, P5, and stem-loop P4-SL4 (Fig. 2a, b). Notably, the 2′-OH groups of C24 and G55, form symmetrical hydrogen bonds to the opposite non-bridging phosphate oxygens, resulting in involvement of all four non-bridging phosphate oxygens of the two M5-bridged phosphates in interstrand stabilizing interactions (Fig. 2a, b). Consistent with such an alignment, cleavage activity was abolished for the C24dC/G55dG-substituted twister-sister ribozyme

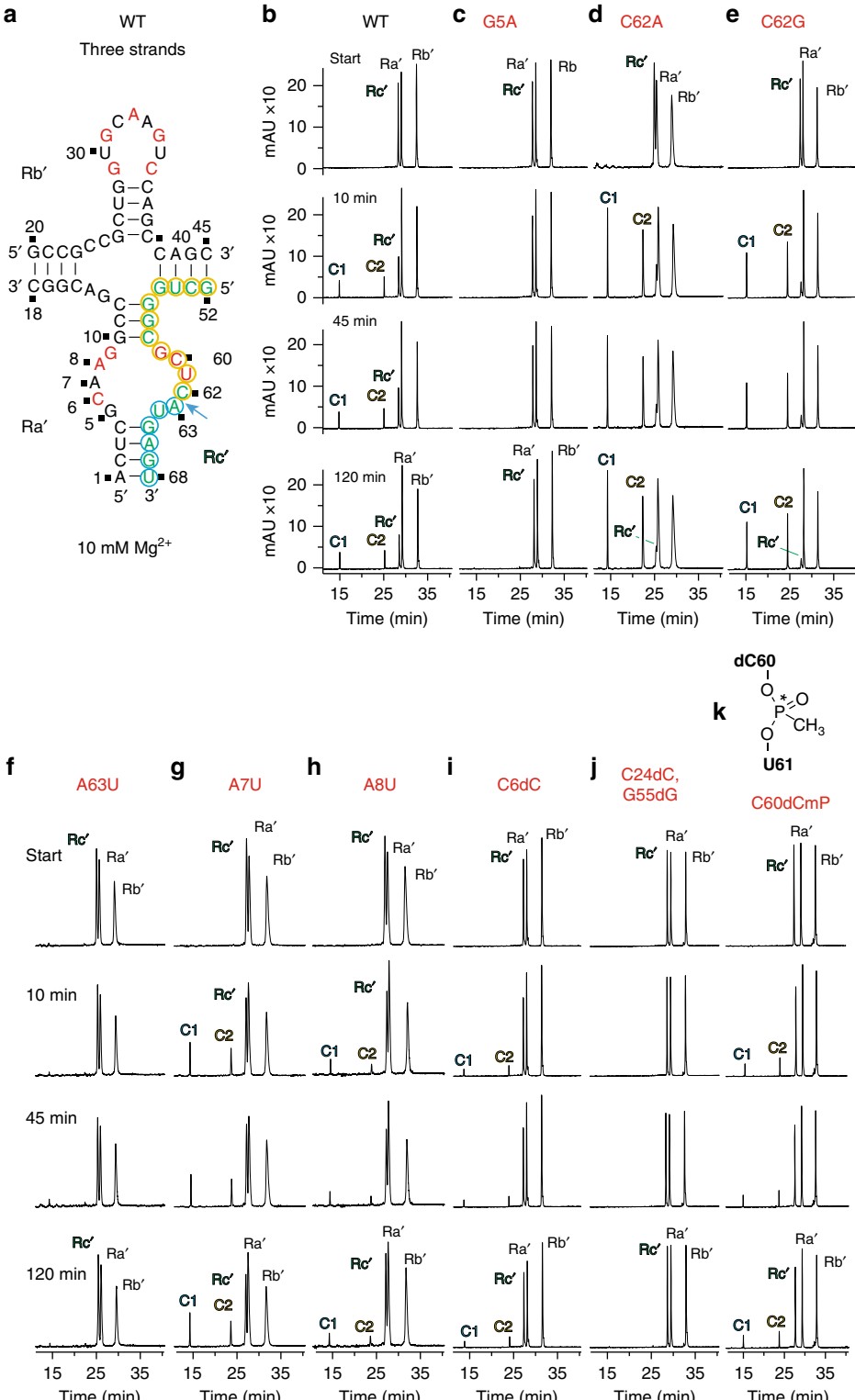

**Fig. 5** Self-cleavage of the three-stranded twister-sister ribozyme. **a** The secondary structure of the three-stranded twister-sister ribozyme used in the cleavage assays. Rc' denotes the 17-nt substrate (green); C1 (blue) and C2 (yellow) denote 6-nt and 11-nt cleavage products. **b–k** HPLC traces following cleavage activity of wild-type ribozyme (**b**) and mutants G5A (**c**), C62A (**d**), C62G (**e**), A63U (**f**), A7U (**g**) and A8U (**h**), C6dC (**i**) and C24dC, G55dG (**j**), and C60dCmP (**k**). Cleavage activity analyzed at 55 µM RNA each strand; 10 mM MgCl₂, 100 mM KCl, 30 mM HEPES, pH 7.5, 23 °C. HPLC conditions: Dionex DNAPac column (4 × 250 mm²), 80 °C, 1 mL min⁻¹, 0–60% buffer B in 45 min. Buffer A: Tris–HCl (25 mM), urea (6 M), pH 8. Buffer B: Tris–HCl (25 mM), urea (6 M), NaClO₄ (0.5 M), pH 8

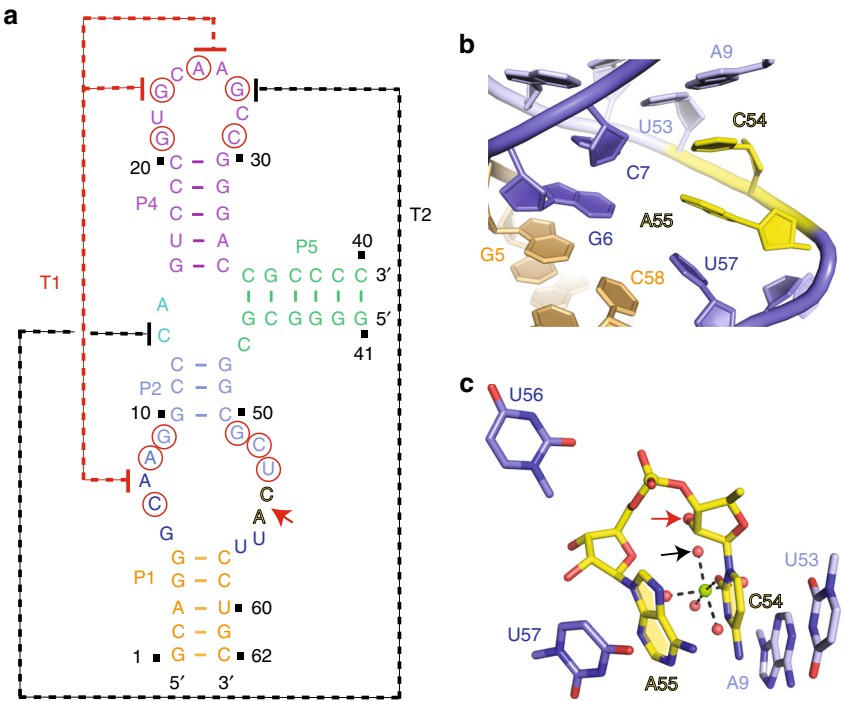

**Fig. 6** Intermolecular contacts and bound Mg$^{2+}$ ions at the C54–A55 cleavage site in the structure (PDB code: 5T5A) of the three-way junctional fold of the twister-sister ribozyme. **a** Schematic of the secondary structure of dC54-containing three-way junctional twister-sister ribozyme (PDB code: 5T5A). All highly conserved residues were labeled with red circles. The long-distance interactions are shown with dash line labeled as T1 (in red) and T2 (in black). The cleavage site is shown with a red arrow. **b** Expanded view of the cleavage site. C54 is stacked with A55, and this stacked pair is sandwiched between bases A9 and U57. **c** An inner-sphere water of a hydrated Mg$^{2+}$ is within hydrogen-bonding distance of the modeled 2′-OH C54. The enol tautomer form of U56 may form a hydrogen bond with the *pro-S$_P$* non-bridging phosphate oxygen of the cleavage step

(Fig. 5j). Similarly, a trio of sequentially aligned hydrated Mg$^{2+}$ cations (M3, M1, and M4) involved primarily in non-bridging phosphate oxygen coordination form a stabilizing network bridging opposingly positioned G5–C6 and C58–G59–C60–U61 segments proximal to the C62–A63 cleavage site (Fig. 3e).

Nucleotides A7, G35, U36, and G59 are looped out of the stacked helical-rich architecture of the twister-sister ribozyme. Nevertheless, they are all involved in long-range pairing and stacking interactions stabilizing and interdigitating together the fold of the twister-sister ribozyme. Thus, extruded A7 intercalates between A33 and A34 (Fig. 2c), and in addition, forms base–sugar and base–base hydrogen bonds to A33 and G31 (Fig. 2d). Similarly, extruded G35 and G59 form major groove aligned G35•(C12–C56) (Fig. 2f) and G59•(A8–U61) base triples (Supplementary Fig. 3c). In addition, U36 forms extensive interactions with G59 and the A8–G9 step through two hydrated Mg$^{2+}$ cations M6 and M7 (as shown in Fig. 2g), which were confirmed by the anomalous signal collected with Mn$^{2+}$-soaked crystals (Supplementary Fig. 4f, g). Interestingly, one metal designated as M8, that is coordinated to N7 of G35 has strong anomalous signal in the Mn$^{2+}$-soaked crystals (Supplementary Fig. 4f, g), whereas its counterpart is not observed in the native crystal, which may explain why the three-stranded native twister-ribozyme construct has better cleavage activity with Mn$^{2+}$ than Mg$^{2+}$ under 2 mM divalent cation conditions (Supplementary Fig. 1c, d).

The non-conserved A7 residue can be substituted by U without much loss of activity (Fig. 5g). A greater impact on activity is observed for the replacement of the highly conserved A8 by U, which reduces the extent of cleavage but not completely abolishes it (Fig. 5h). This finding is consistent with the possibility of forming a U8•U61 non-canonical base pair, without disturbing

the interaction between G59 and U61 of the base triple (Supplementary Fig. 3c).

We have also monitored the importance of mutating C62 and A63 at the C62–A63 cleavage step in the twister-sister ribozyme. Replacement of flexible C62 by U, A, or G retains cleavage activity (Figs. 4g–i and 5d, e), whereas replacement of anchored A63 by U results in complete loss in activity (Fig. 5f). It appears that the A63U mutation most likely disrupts both the stacking and hydrogen-bonding interactions that anchor A63 within the fold of the twister-sister ribozyme. Furthermore, we probed the *trans* sugar edge-Hoogsteen C6•A63 pair (Supplementary Fig. 3b) by deletion of the 2′-OH of C6; indeed, the C6dC mutant exhibited a reduced extent of cleavage (Fig. 5i).

We made G5 mutations, given that N1H of G5 is directed to the *pro-S$_P$* non-bridging phosphate oxygen at the cleavage step (Fig. 3c, d) and all its ring heteroatoms are involved in hydrogen-bonding interactions (Fig. 3e). Notably, the G5A mutation results in very minor cleavage (Figs. 4c and 5c), suggesting that G5, through its N1H amide functionality (would be N1 imino in adenine), might have an important role for the cleavage chemistry although its direct participation in catalysis remains to be proven, given that it is not absolutely conserved. Similarly, replacement of purine G5 by pyrimidines U or C results in nearly complete loss of activity (Fig. 4d, e). By contrast, the G5I mutant shows cleavage activity (Fig. 4f) comparable to wild-type, implying that disruption of the hydrogen-bonding potential of the 2-amino group of G5 has no effect on cleavage activity.

Earlier studies established that the cleavage rate of the twister-sister ribozyme increased on raising the pH from 5 to 7 before plateauing at neutral pH and also exhibited a steep dependence on Mg$^{2+}$ concentration before plateauing out at 1 mM Mg$^{2+}$ concentration[4]. In the absence of Mg$^{2+}$, no cleavage was detectable in the presence of Co(NH$_3$)$_6$$^{3+}$, nor was cleavage observed

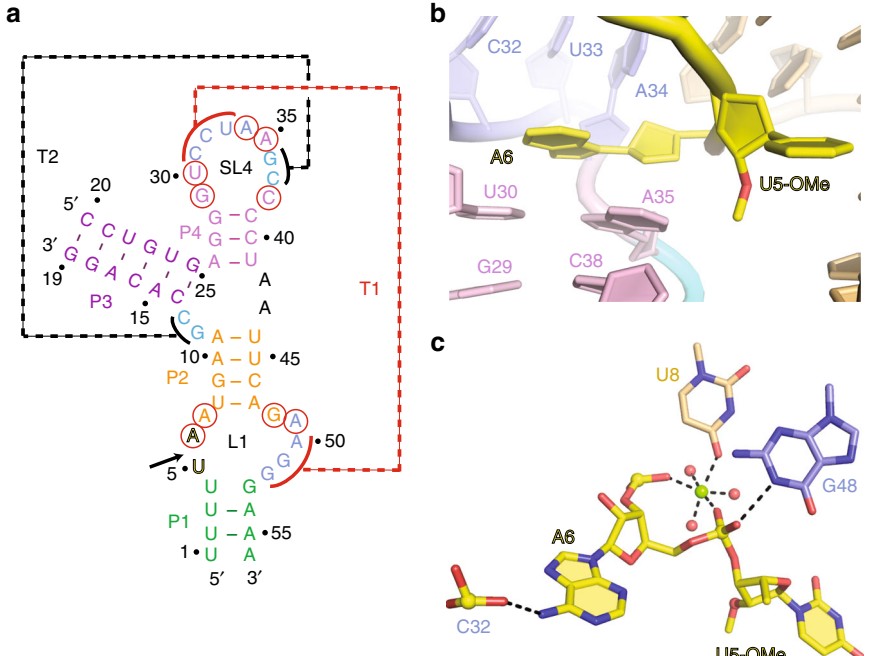

**Fig. 7** Intermolecular contacts and bound metal ions at the U6–A7 cleavage site in the structure of the *env22* twister ribozyme (PDB code: 5DUN). **a** Schematic of the secondary fold of the *env22* twister ribozyme. The sequence is color-coded according to helical segments observed in the tertiary structure. Highly conserved residues are labeled by red circles. Long-distance pseudoknot interactions observed in the tertiary structure are labeled with dashed lines named T1 (in red) and T2 (in black). **b** Bases U5 (modified with a 2′-OCH₃ substitution) and A6 at the cleavage site of the *env22* twister ribozyme are splayed apart, with U5 directed outwards and A6 directed inwards into the ribozyme fold. A6 is stacked with the A35•U30 base pair. **c** The non-bridging phosphate oxygens of the U5–A6 scissile phosphate form hydrogen bonds to the N1H of G48 and directly coordinated to a hydrated Mg$^{2+}$ cation

for monovalent cations (except Li$^+$ at very high concentrations)[4]. Hydrated Mg$^{2+}$ cation M1, with an inner-sphere-coordinated water donating a hydrogen bond to the *pro-R*$_P$ non-bridging oxygen of the scissile phosphate (Fig. 3c, e), is anchored in place through direct coordination to three non-bridging phosphate oxygens. These are the *pro-R*$_P$ oxygen of U61 and the *pro-R*$_P$ oxygen of C60, and also the *pro-S*$_P$ oxygen of G50′ from a symmetrical molecule. To learn more about the potential relevance of the M1 ion from a mechanistic perspective, we synthesized RNA with a methylphosphonate backbone unit between C60 and U61 (C60dCmP) that is expected to interfere with the direct coordinations of the Mg$^{2+}$ ions M1 and M4 that we see in the crystal structure. Despite this interference, we observed that cleavage was only slightly impaired in comparison to the wild-type activity (Fig. 5b, k). We must point out that we used the diastereomeric mixture of methylphosphonate RNA that accounts for maximal 50% disruption of the M1 and M4 coordinations, respectively. The observed activity therefore might arise from the remaining occupied M1 and M4 sites. Clearly, further thorough experiments that utilize diastereomerically pure methylphosphonate RNAs are needed in the future to give more precise answers on the putative role of hydrated M1 in catalysis. The hydrated octahedrally coordinated Mg$^{2+}$ cation M2 is positioned to form an inner-sphere coordination to the modeled 2′-O of C62 at the cleavage step (Fig. 3c, d). There are no base or sugar residues within hydrogen bond distance of the 2′-O of C62 and hence hydrated M2 could be a potential candidate for activation of the 2′-OH of C62, as suggested previously[17].

We did not observe an inline alignment (2′-O to P-O5′ angle of 71°) at the C62–A63 cleavage step in the structure of the twister-sister ribozyme, unlike the near inline alignments observed in our published structure of the twister[9] and pistol[15] ribozymes. This could in part reflect contributions from the flexible extruded

alignment of C62, thereby implying that C62, rather than anchored A63, would need to reorient its conformation to adopt an inline alignment at the C62–A63 cleavage site in the transition-state conformation.

We compare our structure of a twister-sister ribozyme containing a four-way junctional fold (Fig. 1a), with its recently reported structural counterpart from another laboratory containing a three-way junctional fold (Fig. 6a)[17]. These structures are superpositioned with two alternate alignments shown in stereo in Supplementary Fig. 5a and b. A comparative analysis of the global architectures confirmed that the long-range interactions that define the tertiary fold of the twister-sister ribozyme are in most aspects common to both structures. The involved long-distance interaction named T1 (Figs. 1a and 6a) is almost the same, whereas the T2 long-distance interaction (Figs. 1a and 6a) is different between the two structures of the twister-sister ribozyme. In our four-way junctional long-range structure, both the two extruded bases G35 and U36 are involved in long-range interaction labeled T2 (Figs. 1a and 2e–g), with G35 forming a base triple with the major groove edge of the G56–C12 pair (Fig. 2f) and U36 forms extensive hydrogen-bonding interactions with highly conserved base G59, and the non-bridging phosphate oxygens of A8 and G9 through two hydrated Mg$^{2+}$ ions (Fig. 2g). By contrast, in the published three-way junctional structure of the twister-sister ribozyme, only one base-labeled G27 is involved in the interaction labeled T2, where it forms a Watson–Crick G27–C14 pair with C14 from the junctional region (Fig. 6a)[4, 17].

The alignments within the catalytic pocket of the twister-sister ribozyme for cleavage at the C–A step in our four-way junctional structure of the twister-sister ribozyme shown in Fig. 3b, can be compared with its counterpart in the published three-way junctional structure of the twister-sister ribozyme[17] shown in Fig. 6b and c. We note some similarities between these two structures of

the twister-sister ribozyme. For instance, the modeled 2′-OH of C at the cleavage site is within hydrogen-bonding distance of an inner-sphere water of a hydrated $Mg^{2+}$ cation in both structures (Figs. 3c, d and 6c).

By contrast, we note significant differences in orientation as reflected in the splayed-apart alignment of the C–A step, with C adopting flexible and A adopting anchored alignments in our structure (Fig. 3b–d), in contrast to the unanticipated stacked alignment of the C–A step, with C adopting anchored and A adopting flexible alignments in the published structure[17] (Fig. 6b, c). The pro-$S_P$ non-bridging phosphate oxygen at the C–A step is hydrogen bonded to N1H of G5 in our four-way junctional (4-wj) structure (Fig. 3c–e), whereas hydrogen bonding to U56, which is directed towards the pro-$S_P$ non-bridging phosphate oxygen, could only form for the enol tautomer of U56 in the three-way junctional (3-wj) structure[17] (Fig. 6c). In addition, the pro-$R_P$ non-bridging oxygen of the C–A scissile phosphate forms a hydrogen bond with an inner-sphere water of hydrated $Mg^{2+}$ cation M1 in 4-wj structure (Fig. 3c, e), whereas no such hydrated $Mg^{2+}$ site was observed in the 3-wj structure[17] (Fig. 6c). Finally, off-line cleavage alignments are observed both in the 4-wj structure (O2′ P–O5′ angle of 71°) and in the 3-wj structure (67°)[17] for the pre-catalytic state of the twister-sister ribozyme.

These significant differences in the alignments within the catalytic pocket between the four-way (our structure) and three-way[17] junctional folds of the twister-sister ribozyme were unexpected and imply that secondary structure and/or crystallographic packing can influence the topology of the catalytic pocket in these pre-catalytic conformations, without impacting significantly on the global tertiary fold of the ribozyme. In particular, we point out that the one nucleotide larger L1 segment of the three-way junctional (Fig. 6a) compared to the four-way junctional (Fig. 1a) twister-sister ribozyme may contribute to the structural differences at the cleavage site.

In the following, we compare our structures of the twister (Fig. 7)[9, 11] and twister-sister (Fig. 3) (this study) ribozymes and point out alignment similarities within the catalytic pocket of both ribozymes. The catalytic pocket alignments at the U–A cleavage step (U 2′-OH replaced by U 2′-OCH₃) for our 2.6 Å structure of the twister ribozyme[11] shown in Fig. 7c is compared with that at the C-A cleavage step (C 2′-OH replaced by C-2′H) in our 2 Å structure of the twister-sister ribozyme (this study) shown in Fig. 3c and d. Long-range loop–loop interactions involving conserved residues define the tertiary folds and catalytic pocket alignments in twister-sister (Supplementary Fig. 6a, b) and twister (Supplementary Fig. 6c, d) structures. Notably, the pyrimidine–purine step is splayed apart, with the pyrimidine directed outwards and adopting a flexible alignment, whereas the purine is directed inwards and anchored in position through base-stacking interactions in both ribozyme structures. The N1H of a guanine is hydrogen bonded to one of the non-bridging phosphate oxygens (pro-$R_P$ in twister (Fig. 7c) and pro-$S_P$ in twister-sister ribozymes (Fig. 3c, d)), whereas a hydrated $Mg^{2+}$ is coordinated to the other non-bridging phosphate oxygen (direct coordination to pro-$S_P$ in twister (Fig. 7c) and inner-sphere water-mediated coordination to pro-$R_P$ in twister-sister (Fig. 3c, d)).

The outwardly pointing pyrimidine base of the pyrimidine–purine step at the cleavage site adopts a flexile alignment in our structures of the twister and twister-sister ribozymes and can be replaced by adenine and guanine (as well as cytosine and uracil, respectively) in both ribozymes, thereby retaining high cleavage activity[9]. This implies that the positioning of the pyrimidine nucleoside in the pre-catalytic state structures may not reflect its alignment in the transition state, in contrast to

the positioning of the purine nucleoside, whose alignment is stabilized by stacking interactions, and both non-bridging oxygens of the scissile phosphate, which are anchored in place through hydrogen bonding, in the pre-catalytic structures of both ribozymes. Indeed, replacement of this adenine by another base in both ribozymes results in loss of cleavage activity[9]. It is not clear at this time as to how much weight should be placed on the observation of a hydrated $Mg^{2+}$ (M2) whose inner shell water is positioned to hydrogen bond with the modeled 2′-OH of C62 (given the flexibility of the pyrimidine base) in our structure of the twister-sister ribozyme (Fig. 3c, d), given that a hydrated $Mg^{2+}$ cation was not observed in the same position in our structure of the twister ribozyme (Fig. 7c).

It is also not yet clear as to how much weight should be placed on the $Mg^{2+}$ (M1) whose inner shell water is positioned to hydrogen bond with the scissile phosphate oxygen (Fig. 3c, d) that we see for our structure of the twister-sister ribozyme. For twister, we found direct coordination of $Mg^{2+}$ to the scissile phosphate (Fig. 7c); this interaction was probed with enantiomerically pure thiophosphate substrates and resulted in an only about twofold thio rescue (with $Mn^{2+}$ or $Cd^{2+}$)[11], suggesting that divalent metal ions are likely to be minor contributors to the chemical mechanism despite their possible direct interaction in solution[18, 19]. For twister-sister where we only observe the scissile phosphate interacting via a $H_2O$ that is inner-sphere coordinated to $Mg^{2+}$ (Fig. 3c, d), rescue experiments with scissile thiophosphates might encounter limitations. In our view, the above listed similarities within catalytic pocket alignments centered about the to-be-cleaved phosphate group in our structures are supportive of the designations of twister[3] and twister-sister (this study) as related ribozymes[4].

Notably, of previously characterized small self-cleaving ribozymes, the HDV (hepatitis delta virus) ribozyme employs a combination of nucleobase and metal-ion catalysis[20–22], the GlmS ribozyme uses a cofactor (glucosamine-6-phosphate) for catalysis[23–25], and others including hairpin[26, 27], hammerhead[28–30], and VS ribozymes[31] mainly utilize nucleobases as catalytic groups[32]. With the now available structures of the twister-sister ribozyme, it appears likely that this ribozyme reveals additional facets of RNA catalysis for phoshodiester cleavage. How twister-sister employs and combines the main catalytic strategies ($\alpha$, $\beta$, $\gamma$, $\delta$ catalysis)[33], however, is not clear and warrants further investigations. X-ray crystallography has played a key role in defining the overall folding topology of small self-cleaving ribozymes, thereby allowing selective mutation studies on catalytic pocket residues to evaluate their impact on cleavage chemistry. Nevertheless, it should be noted that to prevent cleavage chemistry during crystallization of the ribozymes, the 2′-OH of the nucleotide preceding the cleavage step is generally replaced by 2′-H or 2′-OCH₃, with the potential for such modifications to perturb alignments at the cleavage site.

Further progress towards an improved understanding of mechanistic aspects of cleavage chemistry will depend on moving forward from available structures of pre-catalytic states of the newly identified self-cleaving ribozymes that generally exhibit significant departures from inline alignments at the cleavage site, to structures of transition-state-like vanadate complexes, similar to those reported previously for the hairpin[34] and hammerhead[35] ribozymes, to get a more complete and relevant overview of the potential diversity of catalytic mechanisms adopted by small self-cleaving ribozymes.

## Methods

**Solid-phase synthesis of oligonucleotides.** All oligonucleotides were synthesized on a ABI 392 Nucleic Acid Synthesizer using 2'-O-TOM standard RNA nucleoside phosphoramidite building blocks (ChemGenes) and polystyrene support (GE

Healthcare, Primer Support 5G, 300 μmol per g; PS 200). 5'-O-(4,4'-dimethoxy-trityl)-2'-deoxycytidine phosphoramidite was purchased from ChemGenes.Reaction conditions: detritylation (80s) with dichloroacetic acid/1,2-dichloroethane (4/96); coupling (2.0 min) with phosphoramidites/acetonitrile (0.1 M x 130 μL) and benzylthiotetrazole/acetonitrile (0.3 M × 360 μL); capping (3 × 0.4 min, Cap A/Cap B = 1/1) with Cap A: 4-(dimethylamino)pyridine in acetonitrile (0.5 M) and Cap B: Ac$_2$O/sym-collidine/acetonitrile (2/3/5); oxidation (1.0 min) with I$_2$ (20 mM) in THF/pyridine/H$_2$O (35/10/5). The solutions of amidites and tetrazole, and acetonitrile were dried over activated molecular sieves (3 Å) overnight.

**Deprotection of oligonucleotides.** The solid support was reacted with MeNH$_2$ in EtOH (33%, 0.5 mL) and MeNH$_2$ in water (40%, 0.5 mL) for 7 h at room temperature. The supernatant was removed from and the solid support was washed 3× with THF/water (1/1, v/v). The combined supernatant and the washings were evaporated to dryness. The resulting residue was treated with tetrabutylammonium fluoride trihydrate (TBAF·3H$_2$O) in THF (1 M, 1 mL) at 37 °C overnight to remove the 2'-O-silyl protecting groups. The reaction was stopped by the addition of triethylammonium acetate (TEAA) (1 M, pH 7.4, 1 mL). The volume of the solution was reduced and the solution was desalted using a size exclusion column (GE Healthcare, HiPrep 26/10 Desalting; 2.6 × 10 cm; Sephadex G25) eluting with H$_2$O, the collected fraction was evaporated to dryness and dissolved in 1 ml H$_2$O. The crude RNA after deprotection was analyzed by anion-exchange chromatography on a Dionex DNAPac PA-100 column (4 mm × 250 mm) at 80 °C. Flow rate: 1 mL min$^{-1}$, eluant A: 25 mM Tris·HCl (pH 8.0), 6 M urea; eluant B: 25 mM Tris·HCl (pH 8.0), 0.5 M NaClO$_4$, 6 M urea; gradient: 0–60% B in A within 45 min, UV detection at 260 nm.

**Purification of RNA.** The crude deprotected RNA was purified on a semi-preparative Dionex DNAPac PA-100 column (9 mm × 250 mm) at 80 °C with flow rate 2 ml/min. RNA containing fractions were loaded on a C18 SepPak Plus cartridge (Waters/Millipore), washed with 0.1–0.15 M (Et$_3$NH)$^+$HCO$_3^-$, H$_2$O and eluted with H$_2$O/CH$_3$CN (1/1). RNA containing fractions were lyophilized. The quality of purified RNA was analyzed by anion-exchange chromatography (conditions as for crude RNA; see above). The molecular weight of the RNA was confirmed by LC-ESI mass spectrometry and the yields were determined by UV photometrical analysis of oligonucleotide solutions.

**Mass spectroscopy of RNA.** The RNA was analyzed on a Finnigan LCQ Advantage MAX ion trap instrumentation connected to an Amersham Ettan micro LC system, in the negative ion mode with a potential of -4 kV applied to the spray needle. LC: Sample (200 pmol RNA dissolved in 30 μL of 20 mM EDTA solution; average injection volume: 30 μL); column (Waters XTerraMS, C18 2.5 μm; 1.0 × 50 mm) at 21 °C; flow rate: 30 μL min$^{-1}$; eluant A: 8.6 mM TEA, 100 mM 1,1,1,3,3,3-hexafluoroisopropanol in H$_2$O (pH 8.0); eluant B: methanol; gradient: 0–100% B in A within 30 min; UV detection at 254 nm.

**Crystallization.** We prepared the sample for crystallization by annealing the two strands of chemically synthesized and purified twister-sister ribozyme at 65 for 5 min with a final concentration of 0.4 mM in a buffer containing 50 mM Na-HEPES, pH 6.8, 50 mM NaCl, and 5 mM MgCl$_2$, followed by incubation at room temperature for 5 min, and then cooling on ice for 30 min before setting up crystallization trials.

The crystals of the four-way twister-sister ribozyme were grown by the sitting-drop vapor diffusion method at 20 °C over 7–10 days. We mixed 0.2 μL of the RNA sample (the concentration is 0.4 mM) with an equimolar volume ratio of the reservoir solution. The reservoir solution was composed of 0.1 M NaOAc, pH 5.2, 0.15 M CaCl$_2$, and 18–22% iso-propanol or MPD. The typical crystal dimensions were around 0.05–0.1 mm in a cuboid shape. For data collection, crystals were quickly transferred into a cryoprotectant solution containing 0.1 M NaOAc, pH 5.2, 0.15 M CaCl$_2$ and 30% MPD and flash-frozen in liquid nitrogen. For Ir(NH$_3$)$_6^{3+}$ and Mn$^{2+}$ soaking experiments, crystals were transferred into the crystallization solution containing 0.1 M NaOAc, pH 5.2, 0.15 M CaCl$_2$, and 18–22% MPD and supplemented with 20 mM Ir(NH$_3$)$_6^{3+}$Cl$_3$ or 50–100 mM MnCl$_2$ at 4 °C for 24 h.

**X-ray data collection and refinement.** All X-ray diffraction data were collected at 100K with the beamlines BL-17U1[39], BL-17B, and BL-19U1 at the Shanghai Synchrotron Radiation Facility (SSRF) and processed with HKL2000 (HKL research). The space group was P2$_1$2$_1$2. The phases of the twister-sister ribozyme were solved with the Ir(NH$_3$)$_6^{3+}$ single wavelength anomalous dispersion method. We used HKL2Map[36] to locate the positions of Ir(NH$_3$)$_6^{3+}$ ions. The correlation coefficients output by SHELXD were 37.7 (all data) and 23.5 (weak data) (PAT-FOM of 10.3). Then we used programs SOLVE and RESOLVE in PHENIX Autosol[37] to solve the phases and do the electron density modification (The modification electron density map (contoured at 1σ) was overlaid with the final refined model in Supplementary Fig. 7.). Then, the model was initially built with the program PHENIX Autobuild[37] and then manually built and adjusted using the program Coot[38]. After that, it was used as the initial model to perform molecular replacement on the 2 Å native dataset to solve the native twister-ribozyme

## Table 1 Crystallographic statistics for the four-way junctional twister-sister ribozyme

| Crystal | Native | [Ir(NH$_3$)$_6$]$^{3+}$ soak | Mn$^{2+}$ soak |
|---|---|---|---|
| Data collection | BL-19U | BL-17U | BL-17B |
| Space group | P2$_1$2$_1$2 | P2$_1$2$_1$2 | P2$_1$2$_1$2 |
| Cell dimensions | | | |
| $a, b, c$ (Å) | 91.2, 109.3, 41.7 | 91.3, 109.5, 41.7 | 91.2, 109, 41.8 |
| $\alpha, \beta, \gamma$ (°) | 90, 90, 90 | 90, 90, 90 Peak | 90, 90, 120 |
| Wavelength (Å) | 0.9792 | 1.105 | 1.653 |
| Resolution (Å) | 50–2 (2.07–2)$^a$ | 50–2.10 (2.18–2.10) | 50–2.13 (2.21–2.13) |
| $R_{pim}$ | 0.061(1.099) | 0.053(0.524) | 0.053 (0.509) |
| $I/\sigma I$ | 16.4 (1.5) | 26.7 (2.1) | 23.4 (2) |
| Completeness(%) | 99.7 (99.7) | 100 (100) | 100 (99.8) |
| Redundancy | 4.4 (4.2) | 7.3 (5.8) | 7.5 (6.7) |
| Refinement | | | |
| Resolution (Å) | 35.8–2 (2.07–2) | | 42.1–2.13 (2.17–2.13) |
| No. reflections | 28,779 (2786) | | 44,790 (2682) |
| $R_{work}/R_{free}$ | 0.22/0.24 (0.35/0.41) | | 0.20/0.26 (0.32/0.38) |
| No. of atoms | | | |
| RNA | 2904 | | 2904 |
| Cations | 26 | | 22 |
| Water | 163 | | 286 |
| B-factors (Å$^2$) | | | |
| RNA | 54.8 | | 37.8 |
| Cations | 45.9 | | 46.3 |
| Water | 59.4 | | 34.3 |
| R.m.s deviations | | | |
| Bond lengths (Å) | 0.018 | | 0.005 |
| Bond angles (°) | 0.4 | | 1.006 |

$^a$Values for the highest-resolution shell are in parentheses.

structure, followed by refinement of the structure with PHENIX[37] with a final $R_{work}/R_{free}$ 0.22/0.24 (Table 1). The coordinate error estimated with maximum-likelihood method in PHENIX[37] is 0.36 Å. The 2$F_o$–$F_c$ electron density map (contoured at 1σ) was overlaid with the final refined model in Supplementary Fig. 8. Metal ions and their coordinated waters were identified based on 2$F_o$–$F_c$ and $F_o$–$F_c$ maps guided by the coordination geometries. Mg$^{2+}$ binding sites were identified by soaking crystals of the twister-sister ribozyme in 50 mM Mn$^{2+}$-containing solution and collecting the anomalous data at a wavelength of 1.653 Å. The Mn$^{2+}$-soaked structures were refined using the native twister-ribozyme structure as a starting model. We observe metal-O coordination distances of 2 Å characteristic of Mg$^{2+}$ coordination, rather than the longer 2.4 Å distance characteristic of Ca$^{2+}$ coordination, given the context that Ca$^{2+}$ was the predominant divalent cation present in our crystallization buffers. The X-ray data statistics of the native, Ir (NH$_3$)$_6^{3+}$-containing and Mn$^{2+}$-soaked crystals are listed in Table 1. The structure in figures were prepared using program PyMOL (http://www.pymol.org/).

**Cleavage assays.** Aliquots from aqueous millimolar stock solutions of the two or three RNA strands (Ra, Rb; or Ra', Rb, Rc') were mixed and lyophilized. After addition of reaction buffer (30 mM HEPES, pH 7.5, 100 mM KCl, MgCl$_2$ concentrations as indicated) to yield a final concentration of c(RNA) = 55 μM (each strand) in a total volume of 20 μL, the reaction was stopped by the addition of EDTA solution (20 μL; 3 mM) after 1, 10, 45, and 120 min, stored at 4 °C, and subsequently analyzed by anion-exchange HPLC (analytical Dionex DNAPac column) using the conditions as described above. For the two-strand cleavage assay, the reaction mixture was briefly heated to 70 °C (~20 s) immediately after dissolving of the RNA to support proper annealing.

**Data availability.** The atomic coordinates and structure factors have been deposited in the Protein Data Bank under the following accession codes: 5Y85 for dC62-containing four-way junctional twister-sister ribozyme, and 5Y87 for same ribozyme crystals soaked in Mn$^{2+}$ solution. Other data are available from the corresponding authors upon reasonable request.

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

## Acknowledgements

We thank the staff of the BL-17B and BL-19U1 beamlines at the National Facility for Protein Sciences Shanghai (NFPS) and BL-17U1 at SSRF. We thank Hong Wu and Kaiyi Huang of the Life Sciences Institute (LSI), Zhejiang University for their help in some crystallization solution preparations. The research was supported by grants from the Natural Science Foundation of China (91640104 and 31670826), the Fundamental Research Funds for the Central Universities (2017QN81010), the new faculty start-up funds from Zhejiang University and the Thousand Young Talents Plan of China (A.R.), by Austrian Science Fund FWF P27947 (R.M.), and by NIH 1U19CA179564 (DJP) and NIH P30CA008748 (Cancer Center Core grant to Memorial Sloan-Kettering Cancer Center).

## Author contributions

L.Z. undertook all of the crystallographic experiments with the assistance of Y.Z. under the supervision of A.R. A.R. solved all the structures and undertook the structural analysis with the help of M.T., J.M., and D.J.P. E.M. prepared RNA samples and performed cleavage assays under the supervision of R.M. The paper was written jointly by A.R., R.M., and D.J.P. with input from the remaining authors.

## Additional information

**Competing interests:** The authors declare no competing financial interests.

