## [Peer Review file · Nature Communications]

Reviewer #1 (Remarks to the Author):

NCOMMS-17-09857 Structure-based Insights into Self-Cleavage by a Four-way Junctional Twister-Sister Ribozyme by Luqian Zheng, Elisabeth Mairhofer, Ye Zhang, Masha Teplova, Jinbiao Ma, Dinshaw J. Patel, Ronald Micura and Aiming Ren

This paper concerns the structure of the twister-sister ribozyme (in its form with the P3 helix). A new crystal structure of the ribozyme is described.

The structure of the ribozyme

Broadly the overall structure of this ribozyme is essentially identical to that of Liu et al, which lacked P3. This had the coaxial stacking of P4+P5 and the coaxial stacking of P1 through the loop into P2 (but no P3 of course). Moreover the two tertiary interactions were also described. The metal ions were also described in that study. As expected, the additional helix P3 is coaxial with P1 and P2, and its reassuring to see this for real in the structure. Since most of this is confirmatory I would suggest that the authors concentrate on the distinct aspects, i.e. P3 and the consequent four-way helical junction, and just refer to earlier work for the other aspects of the structure.

The mechanism of the ribozyme

This ribozyme has proved to be very resistant to giving up the details of its mechanism so far. This paper essentially makes no mechanistic advance, and the authors draw few hard conclusions on the probable mechanism.

Going through the manuscript section-by-section :

Abstract

1. "eleven conserved spatially separated loop nucleotides are brought into close proximity at the junctional site"

The use of "junctional" would be better reserved for the four-way junction, not the L1-L4 interaction.

2. The last two sentences need revision due to use of inappropriate conclusions about mechanism.

Introduction

1. “Currently, there is an agreement on the overall folding topology of the twister ribozyme, but disagreement between groups as to details of the alignment at the cleavage step and the role of hydrated Mg^{2+} cations to cleavage chemistry⁵.”

I don't think it is appropriate to discuss the twister mechanism here in this way. One has the feeling that the authors have a sub-text here. For an unbiased analysis of this I recommend people read Breaker's excellent commentary (ref 5 above). The authors make no mention of his actual conclusions - it should be said Breaker most definitely does not support Micura's views in this evaluation !

2. “We report here our structure of the twister-sister ribozyme and identify the role of nucleobase and hydrated Mg^{2+} in catalyzing cleavage chemistry.”

The role of base and cation is far from proved.

3. “...highlights similarities in structural alignments in the catalytic pocket, implying a common cleavage mechanism...”

Even if there was no doubt about the structures, similarities in structure in the absence of biochemical data don't imply a common mechanism. The language is too definite, at best it indicates they may share a mechanism.

Results

Structural Alignments at the Four-way Junctional Site

1. “The minor groove base edge of highly conserved residue A33 forms a hydrogen bond with the 2'-OH of G31,...”

This is wrong. N3 bonds to A7, it is N7 that bonds to G31. There is no reason not to mention both as they are drawn in the figure.

2. Last paragraph. The structure of T2 is surprising as by analogy with twister it might have been expected that G35 would pair with A14. As it is, G59 to which it bonds is better than 90% conserved – why not mention this? More surprising is that A14 pairs to C23 and this appears to be conserved (almost all four-way junction sequences have GA and CC at the start of P3) though there is no obvious reason why this would be. U36 is frequently a C (never a purine) in the alignment but there is no recognition of this or consideration of how this might be accommodated. U36 appears to occupy a similar position to C28 in the Liu structure although the Mg ions are bound differently. C28 is conserved in the 3WJ and a U could not be bound in the same way.

Structural Alignments at the C62-A63 Cleavage Step

1. "Both M4 and M1 are also coordinated to groups in a symmetry related molecule in the crystal lattice (coordination not shown in Figure 3d) so as to maintain octahedral geometries for the divalent metal ion sites."

Not sure what is meant here. On the subject of metal ions, the crystals grew in 2.5 mM MgCl₂ and 75 mM CaCl₂ and were then transferred to 150 mM CaCl₂. 100 mM MnCl₂ was soaked in and the positions directly determined but it is not clear whether sites were occupied by Mg²⁺ or Ca²⁺ in the native set, and no density is shown. It would be surprising if all the sites were occupied by Mg²⁺ given the differences in concentration. So in fig. 3d for instance, is the structure based on the Mn²⁺ structure, with the sites assumed to be Mg²⁺ binding sites too, or is there good evidence for them being occupied by Mg²⁺? I know this is a relatively minor point.

Cleavage Assays on Twister-Sister Containing Base and Sugar Modifications

1. SFig1: In 2 mM divalent this is a slow ribozyme but clearly significantly faster in Mn²⁺ than Mg²⁺. Ignoring the published data, this alone should make one wonder whether bonding to a non-bridging oxygen is the whole story. The rate is significantly slower than that measured by Breaker (or Liu for the three-way junction form), which should also give pause for thought. Obviously there has to be some doubt as to whether chemistry is rate limiting.

2. Fig4a-d: The three-piece construct is even slower and requires more Mg²⁺ to get activity. Thus interpretation of rate data requires even more caution.

3. Fig4e-j, SFig5: Some comments :

C62A is clearly faster and gives a greater extent of cleavage. A surprising result given that the nt is almost always a C or U. It could reflect a flexible positioning of the nt, but it may also arise from greater stability with a purine substitution if chemistry is not rate limiting.

A63U appears to be dead. This is not surprising given all the evidence that positioning of A is important and A is highly conserved. This does not advance our knowledge compared to what is already published.

G5A appears to be dead. G is conserved but there are examples of C and A in the alignment although these have a different nucleotide than U on the opposite strand. The change will undoubtedly disrupt the bonding seen in the structure but this does not prove the G participates in catalysis. One might speculate that a normal Watson-Crick pair is formed (A5:U64), completely disrupting the structure of L1.

C6dC shows a reduced extent of cleavage. Like the WT, there appears to be no significant change after 10 min so it may be that the rate of cleavage is unchanged, only the proportion of molecules in an active structure is altered. Thus these data cannot be used to support the importance of the hydrogen bond between A63 and C6.

C24dC,G55dG show a tiny hint of activity after 120 min. This doesn't prove that the structure at the four-way junction is correct but it certainly strongly supports it.

A7U looks unchanged from WT. This is consistent with A, C and U being in the alignment as previously discussed.

A8U appears to be similar to C6dC – a reduced extent of cleavage but not necessarily a reduced rate. This is surprising since A8 is completely conserved, participates in a triple with U61 and G59, and its N3 is very important. O2 of U can plausibly replace N3 of A in accepting a hydrogen bond and probably the true rate is lower.

Discussion

Stabilization of the Junctional Alignment

1. "Such an alignment is validated from cleavage assays on C24dC/G55dG substitutions of the twister-sister ribozyme, which abolished activity (Figure 4j)."

As stated above, it is not validated. The mutant just demonstrates that at least one hydroxyl group is important. Of course it is consistent with it. The structure is a perfect parallel four-way junction which ensures a close approach of the two exchanging strands. There can't be too many examples in the database at this sort of resolution. The authors might look for others – the Mg²⁺ binding might be a common motif.

2. "Similarly, a trio of sequentially aligned hydrated Mg²⁺ cations (M3, M1 and M4) involved primarily in non-bridging phosphate oxygen coordination form a stabilizing network..."

Yes, but M1 and M4 make bonds to an adjacent molecule in the lattice so their relevance is uncertain. And the whole area is likely to be remodeled to some extent.

3. "Interestingly, one metal designated as M8, that is coordinated to N7 of G35 has strong anomalous signal in the Mn²⁺-soaked crystals (Supplementary Figure 4f, g), while its counterpart is not observed in the native crystal, which may explain why the three-stranded native twister-ribozyme construct has better cleavage activity with Mn²⁺ than Mg²⁺ (Figure 4b, c)."

Or it may not be the reason at all ! Please consider the published data.

Impact of Nucleotide Substitutions Lining Junctional Site

1. "Interestingly, A7 can be substituted by U without much loss of activity (Supplementary Figure 5b), suggesting that its reduced stacking propensity is likely compensated by hydrogen bonding to G31."

In what manner? Further bonding would change the orientation of the bases, disrupting the bonding to A33 and destabilizing the loop. But a straight replacement is plausible.

2. "More impact on activity is observed for the replacement of A8 by U, which reduces but not completely abolishes cleavage (Supplementary Figure 5c). This finding is consistent with the possibility of forming a U8•U61 non-canonical base pair, without disturbing the interaction between G59 and U61 of the base triple (Supplementary Figure 3c)."

As argued above, it is not clear to what extent the rate of cleavage is reduced. The second sentence is true !

Impact of Nucleotide Substitutions Lining Cleavage Pocket

1. "...indeed, the C6dC mutant exhibited significantly reduced cleavage (Figure 4i)."

As argued above, the extent of cleavage is reduced but the effect on the rate is not clear and may be negligible. Thus the importance of the C6.A63 pair is uncertain.

2. "Notably, the G5A mutation results in complete loss in cleavage activity (Figure 4h), suggesting that G5, through its N1H position (would be N1 in adenine), plays a key catalytic role in the cleavage chemistry."

As argued above, the disruption is severe so it does not prove it participates in catalysis, merely that an A is very bad.

Role of Mg²⁺ Cations M1 and M2 in Cleavage Chemistry

1. "Hydrated Mg²⁺ cation M1, which forms an inner-sphere coordination to the *pro*-RP non-bridging phosphate oxygen at the cleavage step (Figure 3c), is anchored in place through direct coordination to three non-bridging phosphate oxygens. Thus, it is likely that octahedrally coordinated hydrated Mg²⁺ cation M1 plays a key role in cleavage chemistry."

One of the phosphate oxygen atoms comes from a different molecule! Is the cation even bound in solution?

3. "Hydrated octahedrally coordinated Mg²⁺ cation M2 is positioned to form an inner- sphere coordination to the 2'-O of C62 at the cleavage step (Figure 3c). There are no base or sugar residues within hydrogen bond distance of the 2'-O of C62 and hence M2 could be a potential candidate for activation of the 2'-OH of C62."

Figure 3c does a poor job of showing what could potentially be an important finding. Nor is its positioning and binding clear in Sfig 4.

Structure Reflects a Pre-catalytic Conformation of the Twister-Sister Ribozyme

This observation should have preceded all the section on Mg²⁺ and G5. Clearly C62 or A63 have to move to get inline, and perhaps both. Thus the bonding to the scissile phosphate is, at best, of uncertain relevance. Add to this the bonding by metal ions to another molecule and the non-conserved nature of G5 and you rally have learnt very little about the active site.

Structural Similarities Between Twister and Twister-Sister Ribozymes

This whole section is of rather questionable value. The relevance of the bonding of Mg^{2+} and G5 for TS is doubtful, and I would be surprised if either proved to be correct. These authors structure of twister either had a seriously disrupted P1 helix, or no helix, and comparison is not recommended. The conclusion that they are likely to use a common mechanism is unwarranted.

Reviewers' comments:

Reviewer #2 (Remarks to the Author):

The manuscript by Zheng et al describes the crystal structure of a Twister-Sister (TS) ribozyme. Earlier this year, Liu et al reported a structure of a TS ribozyme with a three-way junction architecture (Nature Chem Biol 13:508). The structure now reported is of a four-way junction TS ribozyme. While the two structures have many similarities, the active site architecture is surprisingly different. This difference (which may suggest conformational flexibility or something else), and the different overall fold (3-way vs. 4-way junction) warrant consideration for publication of this manuscript. Overall the work seems technically sound, with some weaknesses in presentation.

In no particular order:

1. Abstract, I would simply say "in a previously published structure of the twister ribozyme" "... our published structure ..." reads unprofessional
2. Abstract, is "imply common catalytic strategies" a bit too strong? What about "suggest"
3. Page 7, line -4 and elsewhere, the correct nomenclature is syn and anti conformations. Not "alignments"
4. Page 12, first sentence of "Structural Comparison ..." section has the clause "form another laboratory" twice.
5. Discussion. It would make the significance of the work more clear to the general reader if it is mentioned that of previously characterized "small" self-cleaving ribozymes, the HDV employs a combination of nucleobase and metal-ion catalysis, the glmS can be converted into a metalloenzyme by a single point mutation, and that others (hairpin, hammerhead, VS) appear to employ nucleobases as catalytic groups; thus, the TS shows another way in which RNA catalysis of the same reaction can happen.
6. Methods, page 16, what was the concentration of the RNA in the annealing solution? What was its concentration in the solution that was mixed 1:1 with reservoir for crystallization? What were typical dimensions of the crystals? Their morphology?
7. Regarding the "native" dataset, since there was quite a bit (150 mM) of Ca²⁺ present, did the authors inspect the anomalous difference Fourier for signs of Ca²⁺ binding to their imputed Mg²⁺ binding sites?
8. In the crystallographic methods, some indication of the quality of the experimental phases (for instance, phasing power, mean overall figure of merit prior to density modification) needs to be mentioned. Also, was density modification performed? With what program (most likely Resolve, if using PHENIX, but the authors need to say)? Also, what target function was used for refinement (specifically, was it a function with experimental phases)?
9. Density modified (presumably) experimental electron density for the asymmetric unit needs to be shown in supplementary, overlaid on final refined model.
10. The mean precision of the refined crystallographic coordinates (Luzzati, sigmaA, etc) needs to be stated in methods.
11. A side-by-side figure comparing 3D cartoon representations of the 3-way and 4-way TS structures should be included in supplementary

12. S Table 1, it is mean I over mean sigma of I, which is represented $\langle I \rangle / \langle \sigma(I) \rangle$ (with the s being a lower-case Greek sigma). Also, B-factors have units (angstroms squared). The refinement section should also have the highest-resolution shell values for resolution, number of reflections, and R values.

13. The reference number 17 is incomplete.

14. Figure 5b should be in a representation where the 2' substituents of the riboses are visible

Reviewer #3 (Remarks to the Author):

Zheng et al. present the crystal structure of a twister sister ribozyme and present data to support the hypothesis that it utilizes metal ions in its catalytic mechanism. The crystal structure determination is of high quality and resolution. Zheng et al. compare this structure to an existing crystal structure of the twister sister ribozyme from David Lilley's group. The twister sister ribozyme structure from the Lilley group led to the hypothesis that this specific class of ribozymes are metalloenzymes. In contrast, the twister ribozyme was previously shown to employ nucleobases in the catalytic mechanism with no requirement for a metal ion. Zheng et al. aim to resolve this discrepancy by solving the structure of a different twister sister ribozyme. However, the conclusions derived from this analysis are relatively weak and uncertain. In my opinion, there was no clear biochemical or structural evidence for the involvement of a metal ion in the catalytic mechanism.

Major Points:

M1 is hypothesized to be the catalytic metal ion, however the binding sites for M1 in both twister-sister structures is not conserved. In the Lilley structure, it is forming a nucleobase interaction, whereas Zheng et al. observe inner shell coordinations to the phosphate oxygen atoms (of nucleotides G50 and C60). In addition, the location of the metal ion is different in both structures with respect to the cleavage site. Active site architecture is typically conserved between different enzymes employing the same mechanism. This active site architecture should be especially conserved between twister sister ribozymes since they are in the same class. This lack of conservation does not support the metalloenzyme hypothesis. In addition, the authors did not present definitive biochemical analyses to support this assertion. I would suggest phosphorothioate substitution and recovery to support the hypothesis that M1 is playing catalytic roles. I mention M1 since it is the only one in the active site forming inner shell coordination with G50 and C60, whereas M2 only interacts via outer shell coordination and is therefore difficult to probe. There was also no detailed mechanism illustrating the possible precise role of M1 in catalysis.

Minor Points

- incorrect use of the word "step" throughout the paper
- the word "likely" is used too often in this paper
- many grammatical mistakes

NCOMMS-17-09857 Structure-based Insights into Self-Cleavage by a Four-way Junctional Twister-Sister Ribozyme by Luqian Zheng, Elisabeth Mairhofer, Ye Zhang, Masha Teplova, Jinbiao Ma, Dinshaw J. Patel, Ronald Micura and Aiming Ren

To address the critiques by the reviewers we have conducted **additional experiments** that were challenging with respect to the chemical synthesis of large-size RNAs and the type of modifications needed (methylphosphonates). Based on these RNAs, **eleven** new cleavage assays of the **2-strand** twister-sister constructs (G5A, G5U, G5C, G5I, C62U, C62A and C62G; Mg²⁺, Mn²⁺, Ca²⁺, Sr²⁺) and **two** new cleavage assays of the **3-strand** constructs (C62G and C60dCmP) are included in the revised manuscript and provide an additional experimental basis for our rebuttal.

Reviewer 1

This paper concerns the structure of the twister-sister ribozyme (in its form with the P3 helix). A new crystal structure of the ribozyme is described.

We appreciate that the reviewer acknowledges the novelty of our study.

The structure of the ribozyme

Broadly the overall structure of this ribozyme is essentially identical to that of Liu et al, which lacked P3. This had the coaxial stacking of P4+P5 and the coaxial stacking of P1 through the loop into P2 (but no P3 of course). Moreover the two tertiary interactions were also described. The metal ions were also described in that study. As expected, the additional helix P3 is coaxial with P1 and P2, and its reassuring to see this for real in the structure. Since most of this is confirmatory I would suggest that the authors concentrate on the distinct aspects, i.e. P3 and the consequent four-way helical junction, and just refer to earlier work for the other aspects of the structure.

We strongly dispute this reviewer's misleading comments that our structure of the four-way junctional twister-sister ribozyme is essentially identical to that of the recently reported three-way junctional ribozyme counterpart and that the results are confirmatory. We have made a serious effort to compare the two structures in our manuscript and gone to considerable effort to point out the similarities and differences in alignment both at the global level and at the cleavage step. The main difference at the global level involves the long-range interaction T2, which is very different between the four-way (Figure 1a) and three-way (Figure 6a) junctional structures of the twister-sister ribozyme. More importantly, there are major differences at the C-A cleavage step and alignment of surrounding residues and divalent cations between the four-way (Figure 3c) and three-way (Figure 6c) junctional structures of the twister-sister ribozyme. We observe a splayed-apart alignment at the C-A cleavage step, with C adopting a flexible looped-out alignment and A stacked into the duplex in the four-stranded junctional structure (Figure 3c). By contrast, the bases are stacked rather than splayed-apart at the C-A cleavage step, with A adopting a flexible alignment and C stacked into the duplex in the three-way junctional structure (Figure 6c). Further, the guanine (G5) and hydrated Mg²⁺ cation (M1) that are coordinated to the non-bridging phosphate oxygens of the scissile phosphate in our structure of the four-way junctional twister-sister ribozyme (Figure 3c and 3d) were not observed in the published structure of the three-way junctional ribozyme (Figure 6c).

This is not just our position, but reviewer 2 also makes the following introductory statement.

*“While the two structures have many similarities, the **active site architecture is surprisingly different**. This difference (which may suggest conformational flexibility or*

something else), and the different overall fold (3-way vs. 4-way junction) warrant consideration for publication of this manuscript. Overall the work seems technically sound, with some weaknesses in presentation.”

The mechanism of the ribozyme

This ribozyme has proved to be very resistant to giving up the details of its mechanism so far. This paper essentially makes no mechanistic advance, and the authors draw few hard conclusions on the probable mechanism.

Both our structure of the four-way junctional twister-sister ribozyme and the published structure of the three-way junctional counterpart represent pre-catalytic conformations. Neither structure adopts an in-line alignment at the scissile phosphate required for cleavage chemistry, suggestive of the requirement for structural changes on formation of the transition state conformation. In our four-way junctional ribozyme, C is flexible and A is anchored in place, with the non-bridging phosphate oxygens at the scissile step also anchored in place through hydrogen bonding interactions (Figure 3c). We therefore anticipate that C will most likely undergo a conformational change to generate the transition state conformation. It is our position that it is best to await results related to efforts to generate and solve structures of vanadate transition-state mimics of the twister-sister ribozyme prior to reaching definitive mechanistic conclusions. We have therefore toned down mechanistic claims throughout the manuscript in the revised version (for details, see responses to the individual points below). In addition, we have found further evidence that G5 plays an important role for self-cleavage of this ribozyme (see below).

Going through the manuscript section-by-section:

Abstract

1. “eleven conserved spatially separated loop nucleotides are brought into close proximity at the junctional site”.

The use of "junctional" would be better reserved for the four-way junction, not the L1-L4 interaction.

Changed as suggested.

2. The last two sentences need revision due to use of inappropriate conclusions about mechanism.

We have changed the text towards the end of the Abstract with the following replacement (see below) focused on comparing the structures of the four-way and three-way twister-sister ribozymes.

“Our four-way junctional pre-catalytic structure differs significantly in the alignment at the cleavage step (splayed-apart versus base-stacked) and surrounding residues relative to a recently reported three-way junctional pre-catalytic structure of the twister-sister ribozyme.”

Introduction

1. “Currently, there is an agreement on the overall folding topology of the twister ribozyme, but disagreement between groups as to details of the alignment at the cleavage step and the role of hydrated Mg²⁺ cations to cleavage chemistry 5.”

I don't think it is appropriate to discuss the twister mechanism here in this way. One has the feeling that the authors have a sub-text here. For an unbiased analysis of this I recommend people read Breaker's excellent commentary (ref 5 above). The authors make no mention of

his actual conclusions - it should be said Breaker most definitely does not support Micura's views in this evaluation!

First, we are very aware of Ronald Breaker's commentary (mentioned by us up front as reference 5 and now as reference 6 in our revised manuscript) and our position is that the issues raised in this commentary can only be addressed following determination of structures of vanadate transition state mimics of the twister ribozyme. Our reading of this commentary is that Breaker is concerned about the limitations of pre-catalytic conformations in addressing mechanistic issues and hence is not in a position to either support or dispute Micura's or Lilley's position on this issue.

As suggested, we have deleted this sentence, shortened the paragraph and rephrased to present a balanced evaluation:

"The earliest studies on recently identified small self-cleaving ribozymes focused on the twister ribozyme and included structural⁸⁻¹⁰, mechanistic^{11,12} and computational^{13,14} contributions. The currently available structures of twister ribozymes reflect pre-catalytic conformations that display distinctions at the cleavage site with respect to the attacking 2'-O P-O5' alignment and the presence/absence of Mg²⁺ cations at the scissile phosphate. All structures are in agreement that the pre-catalytic fold of the twister ribozyme, provides conformational flexibility of the pyrimidine nucleoside at the pyrimidine-purine cleavage step in contrast to the anchored purine nucleoside, suggestive of a likely realignment of the pyrimidine and its 2'-OH to achieve in-line alignment in the transition state. Interestingly, this ribozyme can cleave a single nucleoside (5' of the scissile phosphate) with only two-fold reduced rate and despite the inability to form the phylogenetically conserved stem P1¹¹. A single-molecule FRET study on twister folding rationalizes this behavior by revealing that the active site-embracing pseudoknot fold is preserved in the shortened ribozyme and in the cleaved 3'-product RNA that both lack P1¹²."

2. "We report here our structure of the twister-sister ribozyme and identify the role of nucleobase and hydrated Mg²⁺ in catalyzing cleavage chemistry."
The role of base and cation is far from proved.

We toned down the wording to

"We report here the structure of a four-way junctional twister-sister ribozyme and discuss possible/putative roles of selected nucleobases and Mg²⁺ ions that are located at (or close to) the active site for cleavage chemistry."

Importantly, we performed additional mutation experiments that strengthen our original conclusion about the likely involvement of G5 in catalysis. All of the new cleavage assays (G5U, G5C, G5A, and G5I) were set up using the 2-strand twister-sister complex to address the reviewer's concerns with respect to limitations that might arise from rate-limiting assembly of the 3-strand complex (see below).

Additionally, we conducted cleavage experiment on C62 mutants (C62A, C62U, C62G) using the 2-strand complex. Again, the original observation obtained from the assays using 3-strand construct were confirmed. The pronounced cleavage activities observed for all three mutants suggest limited interactions of the nucleobase-62 with the rest of the active site pocket. They support the high relevance of the exposed conformation that we see in the crystal structure, and by extension, also for conformations in solution. Furthermore, we evaluated the C62G mutant also in the 3-strand assay and observed cleavage.

Of note, in the 3-strand assay, C62A as well as C62G reach higher cleavage yields compared to the wild-type while in the 2-strand assay, C62A shows comparable-to-wild-type yield but C62G shows slower cleavage and less yield.

Furthermore, we performed an additional experiment based on methylphosphonate modified RNA that is expected to interfere with the inner sphere coordinations of the Mg^{2+} ions M1 and M4 that we see in the crystal structure. Despite this interference, we observed that cleavage was only slightly impaired in comparison to the wild-type activity. We must point out that we have used the diastereomeric mixture of methylphosphonate RNA that accounts for maximal 50% disruption of the M1 and M4 coordination, respectively. The observed activity therefore might arise from the remaining occupied M1 and M4 sites. Clearly, thorough experiments that utilize diastereomerically pure methylphosphonate RNAs are needed in the future to give more precise answers on the putative role of M1 in catalysis.

We added the corresponding HPLC cleavage assay on the methylphosphonate as new panel to Figure 5k.

3. "...highlights similarities in structural alignments in the catalytic pocket, implying a common cleavage mechanism..."

Even if there was no doubt about the structures, similarities in structure in the absence of biochemical data don't imply a common mechanism. The language is too definite, at best it indicates they may share a mechanism.

We agree and toned down the wording:

"Finally, a comparison between previously reported structures of the twister ribozyme^{9,11} and the structure of the twister-sister ribozyme (this study) highlights similarities in structural alignments in the catalytic pocket that may indicate shared concepts of these two related ribozymes to mediate cleavage by involving a specific guanine–scissile phosphate interaction, a splayed apart conformation of the cleavage site, and involvement of – although to different extent – hydrated Mg^{2+} ions."

Results

Structural Alignments at the Four-way Junctional Site

1. "The minor groove base edge of highly conserved residue A33 forms a hydrogen bond with the 2'-OH of G31,..."

This is wrong. N3 bonds to A7, it is N7 that bonds to G31. There is no reason not to mention both as they are drawn in the figure.

We thank the reviewer for pointing out this error. In the revised manuscript, we now mention both interactions.

2. Last paragraph. The structure of T2 is surprising as by analogy with twister it might have been expected that G35 would pair with A14. As it is, G59 to which it bonds is better than 90% conserved – why not mention this? More surprising is that A14 pairs to C23 and this appears to be conserved (almost all four-way junction sequences have GA and CC at the start of P3) though there is no obvious reason why this would be. U36 is frequently a C (never a purine) in the alignment but there is no recognition of this or consideration of how this might be accommodated. U36 appears to occupy a similar position to C28 in the Liu

structure although the Mg ions are bound differently. C28 is conserved in the 3WJ and a U could not be bound in the same way.

We thank the reviewer for pointing out these details. In the revised manuscript, we now explicitly list the highly conserved residues that are involved in T2 formation and refer to the corresponding Figure panels.

Structural Alignments at the C62-A63 Cleavage Step

1. “Both M4 and M1 are also coordinated to groups in a symmetry related molecule in the crystal lattice (coordination not shown in Figure 3d) so as to maintain octahedral geometries for the divalent metal ion sites.”

Not sure what is meant here. On the subject of metal ions, the crystals grew in 2.5 mM MgCl₂ and 75 mM CaCl₂ and were then transferred to 150 mM CaCl₂. 100 mM MnCl₂ was soaked in and the positions directly determined but it is not clear whether sites were occupied by Mg²⁺ or Ca²⁺ in the native set, and no density is shown. It would be surprising if all the sites were occupied by Mg²⁺ given the differences in concentration. So in fig. 3d for instance, is the structure based on the Mn²⁺ structure, with the sites assumed to be Mg²⁺ binding sites too, or is there good evidence for them being occupied by Mg²⁺? I know this is a relatively minor point.

As shown in Supplementary Figure 4a, there are two molecules of the four-way junctional twister-sister ribozyme packed against each other in the crystal structure. We note that M4 and M1 achieve octahedral alignment through coordination with atoms in both molecules.

The reviewer is correct in that our Mn²⁺ substitution data has identified divalent cation sites in the twister-sister ribozyme and it is important to distinguish between Mg²⁺ and Ca²⁺ sites, especially given the high Ca²⁺ concentrations used in the crystallization trials.

The coordination lengths between divalent metals and water oxygens are normally 2.4 Å for hydrated Ca²⁺ cations and 2.0 Å for hydrated Mg²⁺ cations. We assign the observed divalent cations to Mg²⁺ in the twister-sister ribozyme structure since we observe lengths of 2.0 Å. This statement has now been added to the Methods section.

Cleavage Assays on Twister-Sister Containing Base and Sugar Modifications

1. SFig1: In 2 mM divalent this is a slow ribozyme but clearly significantly faster in Mn²⁺ than Mg²⁺. Ignoring the published data, this alone should make one wonder whether bonding to a non-bridging oxygen is the whole story. The rate is significantly slower than that measured by Breaker (or Liu for the three-way junction form), which should also give pause for thought. Obviously there has to be some doubt as to whether chemistry is rate limiting.

We agree with the reviewer’s notion that the ribozyme is faster in Mn²⁺ than Mg²⁺ as reported by us in Supplementary Figure 1b, d. We also agree that the four-way junctional twister-sister is slower compared to the three-way junctional twister-sister ribozyme that was investigated by Breaker (reference 4 in our revised manuscript) and Lilley (reference 17 in our revised manuscript).

We point out that the four-way junctional twister-sister ribozyme investigated here is in the same rate regime (1 to 5 min⁻¹) as observed for Breaker’s four-way junctional twister-sister ribozymes (reference 4 in our revised manuscript). We believe it is valid to perform base mutation experiments and correlate observed cleavage activities to the three-dimensional arrangement observed for the corresponding nucleosides in the crystal. We agree to discuss

the possible roles they may play in cleavage chemistry with more caution and have revised the manuscript accordingly.

We finally note that the same approach gave important insights into the cleavage mechanism of twister and pistol ribozymes recently; these ribozyme classes exhibit rate regimes (1 to 10 min⁻¹) comparable with the 4-way junctional twister-sister class.

2. Fig4a-d: The three-piece construct is even slower and requires more Mg²⁺ to get activity. Thus interpretation of rate data requires even more caution.

We agree. We therefore repeated and extended the base mutation experiments of the most crucial nucleotides (G5 and C62) in the 2-strand ribozyme set-up. The new results (see below) strengthen the original observations that were obtained based on the 3-strand construct.

3. Fig4e-j, SFig5:

Some comments :

C62A is clearly faster and gives a greater extent of cleavage. A surprising result given that the nt is almost always a C or U. It could reflect a flexible positioning of the nt, but it may also arise from greater stability with a purine substitution if chemistry is not rate limiting.

To support the notion that the C62A mutant reflects flexible positioning (and not greater stability with a purine substitution) we now analyzed the C62A mutation also in the 2-strand construct and found the same reactivity profile as for the wild-type. In additional experiments, C62U also showed high cleavage activity and further supports the notion of flexible positioning for a nucleobase at this position. Furthermore, we evaluated the C62G mutant in both the 2- and 3-strand assay and again observed good and high activity, respectively.

Notably, in the 3-strand assay, C62A and C62G appear of faster rate and clearly give higher yields compared to the wild-type control, while in the 2-strand assay, C62A shows very similar-to-wild-type yield but C62G is slower and gives less yield. At the moment, we have no explanation for this behavior.

A63U appears to be dead. This is not surprising given all the evidence that positioning of A is important and A is highly conserved. This does not advance our knowledge compared to what is already published.

In our opinion, this finding advances knowledge compared to what is already published: A63 is stacked between U64 and A8 in our x-ray structure (not observed for the structure of the three-way junctional twister-sister); the inactivity of the A63U mutant therefore likely corresponds to its reduced stacking propensity.

G5A appears to be dead. G is conserved but there are examples of C and A in the alignment although these have a different nucleotide than U on the opposite strand. The change will undoubtedly disrupt the bonding seen in the structure but this does not prove the G participates in catalysis. One might speculate that a normal Watson-Crick pair is formed (A5:U64), completely disrupting the structure of L1.

To further explore the role of G5, we performed additional experiments, all based on the 2-strand construct. Cleavage was nearly abolished for G5U and G5C mutants, and severely slowed down for G5A. Mutation of G5 to inosine (G5I) resulted in retaining full cleavage activity; this is consistent with I5 retaining the possibility to form the N1-H⁺...O-P hydrogen bond to the scissile phosphate. G5I also provides the characteristic Hoogsteen face of G5

that is involved in water-mediated hydrogen bonding patterns to M1 and M3 (Figure 3e). Since G5I cannot serve to form the native NH₂···O U64 interaction, this interaction might be less important for activity.

We stress that the observed G5 N1-H···O-P hydrogen bond to the scissile phosphate (as well as the corresponding G5 mutants' cleavage activities) are reminiscent to the G48 N1-H···O-P hydrogen bond to the scissile phosphate in the related twister ribozyme (reference 8-10 in our revised manuscript). This may indicate a common mechanistic role of G5 (twister-sister) and G48 (twister).

C6dC shows a reduced extent of cleavage. Like the WT, there appears to be no significant change after 10 min so it may be that the rate of cleavage is unchanged, only the proportion of molecules in an active structure is altered. Thus these data cannot be used to support the importance of the hydrogen bond between A63 and C6.

As suggested by this reviewer, we have toned down the wording in the revised text.

C24dC,G55dG show a tiny hint of activity after 120 min. This doesn't prove that the structure at the four-way junction is correct but it certainly strongly supports it.

We agree that our data strongly supports the proposed alignment at the four-way junction and at the same time are wondering what kind of experiment the reviewer would suggest to "prove" that the structure is correct.

A7U looks unchanged from WT. This is consistent with A, C and U being in the alignment as previously discussed.

We agree.

A8U appears to be similar to C6dC – a reduced extent of cleavage but not necessarily a reduced rate. This is surprising since A8 is completely conserved, participates in a triple with U61 and G59, and its N3 is very important. O2 of U can plausibly replace N3 of A in accepting a hydrogen bond and probably the true rate is lower.

We agree.

Discussion

Stabilization of the Junctional Alignment

1. "Such an alignment is validated from cleavage assays on C24dC/G55dG substitutions of the twister-sister ribozyme, which abolished activity (Figure 4j)."

As stated above, it is not validated. The mutant just demonstrates that at least one hydroxyl group is important. Of course it is consistent with it. The structure is a perfect parallel four-way junction which ensures a close approach of the two exchanging strands. There can't be too many examples in the database at this sort of resolution. The authors might look for others – the Mg²⁺ binding might be a common motif.

We have rephrased the text of the revised version to read

*"**Consistent** with such an alignment, cleavage activity was abolished for the C24dC/G55dG-substituted twister-sister ribozyme (Figure 5j)."*

2. “Similarly, a trio of sequentially aligned hydrated Mg²⁺ cations (M3, M1 and M4) involved primarily in non-bridging phosphate oxygen coordination form a stabilizing network...”

Yes, but M1 and M4 make bonds to an adjacent molecule in the lattice so their relevance is uncertain. And the whole area is likely to be remodeled to some extent.

As stated above, to learn more about the relevance/non-relevance of the M1 ion, we synthesized RNA with a methylphosphonate backbone unit between C60 and U61 (C60dCmP) that is expected to interfere with the inner-sphere coordinations of the Mg²⁺ ions M1 and M4 that we see in the crystal structure. Despite this interference, we observed that cleavage was only slightly impaired in comparison to the wild-type activity. We must point out that we have used the diastereomeric mixture of methylphosphonate RNA that accounts for maximal 50% disruption of the M1 and M4 coordination, respectively. The observed activity therefore might arise from the remaining occupied M1 and M4 sites. Clearly, thorough experiments that utilize diastereomerically pure methylphosphonate RNAs are needed in the future to give more precise answers on the putative role of M1 in catalysis. The above comments have been added to the text of the revised version.

We added the corresponding HPLC cleavage assay as new panel k to Figure 5.

3. “Interestingly, one metal designated as M8, that is coordinated to N7 of G35 has strong anomalous signal in the Mn²⁺-soaked crystals (Supplementary Figure 4f, g), while its counterpart is not observed in the native crystal, which may explain why the three-stranded native twister-ribozyme construct has better cleavage activity with Mn²⁺ than Mg²⁺ (Figure 4b, c).”

Or it may not be the reason at all! Please consider the published data.

We have come up with a potential rationalization and would welcome guidance beyond the statement from the reviewer ‘consider the published data’.

Impact of Nucleotide Substitutions Lining Junctional Site

1. “Interestingly, A7 can be substituted by U without much loss of activity (Supplementary Figure 5b), suggesting that its reduced stacking propensity is likely compensated by hydrogen bonding to G31.”

In what manner? Further bonding would change the orientation of the bases, disrupting the bonding to A33 and destabilizing the loop. But a straight replacement is plausible.

We now just state that the A7U replacement occurs without much loss in activity, without providing an explanation.

2. “More impact on activity is observed for the replacement of A8 by U, which reduces but not completely abolishes cleavage (Supplementary Figure 5c). This finding is consistent with the possibility of forming a U8•U61 non-canonical base pair, without disturbing the interaction between G59 and U61 of the base triple (Supplementary Figure 3c).”

As argued above, it is not clear to what extent the rate of cleavage is reduced. The second sentence is true!

We said “reduces ... cleavage”, and not “reduces the rate of cleavage”. For clarity we explicitly say this in the revised manuscript: “**A greater impact on activity is observed for the**

*replacement of the highly conserved A8 by U, which reduces the **extent of cleavage** but not completely abolishes **it** (Figure 5h)."*

Impact of Nucleotide Substitutions Lining Cleavage Pocket

1. "...indeed, the C6dC mutant exhibited significantly reduced cleavage (Figure 4i)."

As argued above, the extent of cleavage is reduced but the effect on the rate is not clear and may be negligible. Thus the importance of the C6 A63 pair is uncertain.

*For clarity we explicitly say in the revised manuscript: " ..., the C6dC mutant exhibited a reduced **extent of cleavage** (Figure 5i). " We consider this finding notable.*

2. "Notably, the G5A mutation results in complete loss in cleavage activity (Figure 4h), suggesting that G5, through its N1H position (would be N1 in adenine), plays a key catalytic role in the cleavage chemistry."

As argued above, the disruption is severe so it does not prove it participates in catalysis, merely that an A is very bad.

We have addressed this issue in an earlier response listed above, by studying the mutants G5A, G5U, G5C (cleavage activity lost) and G5I (nearly unchanged activity).

We have rephrased and extended this paragraph in the revised manuscript.

Role of Mg²⁺ Cations M1 and M2 in Cleavage Chemistry

1. "Hydrated Mg²⁺ cation M1, which forms an inner-sphere coordination to the pro-RP nonbridging phosphate oxygen at the cleavage step (Figure 3c), is anchored in place through direct coordination to three non-bridging phosphate oxygens. Thus, it is likely that octahedrally coordinated hydrated Mg²⁺ cation M1 plays a key role in cleavage chemistry."

One of the phosphate oxygen atoms comes from a different molecule! Is the cation even bound in solution?

As stated above, to learn more about the relevance/non-relevance of the M1 ion, we synthesized RNA with a methylphosphonate backbone unit between C60 and U61 (C60dCmP) that is expected to interfere with the inner-sphere coordinations of the Mg²⁺ ions (M1 and M4) that we see in the crystal structure. Despite this interference, we observed that cleavage was only slightly impaired in comparison to the wild-type activity. We must point out that we have used the diastereomeric mixture of methylphosphonate RNA that accounts for maximal 50% disruption of the M1 and M4 coordination, respectively. The observed activity therefore might arise from the remaining occupied M1 and M4 sites. Clearly, thorough experiments that utilize diastereomerically pure methylphosphonate RNAs are needed in the future to give more precise answers on the putative role of M1 in catalysis.

Accordingly, we have toned down the wording related to definite involvement of Mg²⁺ ions M1 and M4 in catalysis in the revised manuscript.

3. "Hydrated octahedrally coordinated Mg²⁺ cation M2 is positioned to form an inner-sphere coordination to the 2'-O of C62 at the cleavage step (Figure 3c). There are no base or sugar residues within hydrogen bond distance of the 2'-O of C62 and hence M2 could be a potential candidate for activation of the 2'-OH of C62."

Figure 3c does a poor job of showing what could potentially be an important finding. Nor is its positioning and binding clear in S fig 4.

We have added an alternate view shown in Figure 3d so as to emphasize the role of hydrated M2 as a potential candidate for activation of the 2'-OH of C62.

Structure Reflects a Pre-catalytic Conformation of the Twister-Sister Ribozyme

This observation should have preceded all the section on Mg²⁺ and G5. Clearly C62 or A63 have to move to get in-line, and perhaps both. Thus the bonding to the scissile phosphate is, at best, of uncertain relevance. Add to this the bonding by metal ions to another molecule and the non-conserved nature of G5 and you really have learnt very little about the active site.

As stated above, we have strengthened the important role/relevance of G5 for cleavage activity by having utilized the more robust 2-strand (compared to 3-strand) cleavage assay in combination with additional mutants (U, G, A, and I). Also, the splayed apart conformation observed in our four-way junctional structure but not observed in the published three-way junctional twister-sister is – in our opinion – an important finding about the active site. The relevance of the latter finding has been substantiated by studying additional mutants utilizing the robust 2-strand cleavage assay.

Structural Similarities Between Twister and Twister-Sister Ribozymes

This whole section is of rather questionable value. The relevance of the bonding of Mg²⁺ and G5 for TS is doubtful, and I would be surprised if either proved to be correct. These authors structure of twister either had a seriously disrupted P1 helix, or no helix, and comparison is not recommended. The conclusion that they are likely to use a common mechanism is unwarranted.

We disagree with this comment on rather questionable value because it shows an inherent bias against our published x-ray structure of the twister ribozyme. This reviewer, who is critical about structural features **in the crystal** structure of the twister ribozyme, ignores the fact that “no helix” or “seriously disrupted P1 helix” in the twister ribozyme hardly impacts on the cleavage rate **in solution**. Moreover, this reviewer seems not to be aware that we have NOT published a structure of “no helix” twister ribozyme, but rather our structure contains two base pairs stabilizing the P1 helix.

We are within our right to compare our published crystal structure of the precatalytic state of the twister ribozyme with our current study of the crystal structure of the precatalytic state of the twister-sister ribozyme. In this regard, we make a straight-forward comparison and have toned down the issue of whether they use the same or different catalytic mechanisms, information about which will only emerge following structural studies on transition state mimics.

Reviewer #2

The manuscript by Zheng et al describes the crystal structure of a Twister-Sister (TS) ribozyme. Earlier this year, Liu et al reported a structure of a TS ribozyme with a three-way junction architecture (Nature Chem Biol 13:508). The structure now reported is of a four-way junction TS ribozyme. While the two structures have many similarities, the active site architecture is surprisingly different. This difference (which may suggest conformational flexibility or something else), and the different overall fold (3-way vs. 4-way junction) warrant

consideration for publication of this manuscript. Overall the work seems technically sound, with some weaknesses in presentation.

We thank this reviewer for supporting publication of a revised version of our paper.

In no particular order:

1. Abstract, I would simply say "in a previously published structure of the twister ribozyme" "... our published structure ..." reads unprofessional.

There are two published structures of the twister ribozyme, one from our laboratory and another from David Lilley's laboratory. We used the phrase "our" to distinguish the structure reported by our group from the one reported by the David Lilley group.

2. Abstract, is "imply common catalytic strategies" a bit too strong? What about "suggest"

We have revised the Abstract and no longer discuss common catalytic strategies given that structures of transition state mimics remain to be determined.

3. Page 7, line -4 and elsewhere, the correct nomenclature is syn and anti conformations. Not "alignments"

We have changed "alignments" to "conformations" as requested by this reviewer.

4. Page 12, first sentence of "Structural Comparison ..." section has the clause "form another laboratory" twice.

Corrected in the revised version.

5. Discussion. It would make the significance of the work more clear to the general reader if it is mentioned that of previously characterized "small" self-cleaving ribozymes, the HDV employs a combination of nucleobase and metal-ion catalysis, the glmS can be converted into a metalloenzyme by a single point mutation, and that others (hairpin, hammerhead, VS) appear to employ nucleobases as catalytic groups; thus, the TS shows another way in which RNA catalysis of the same reaction can happen.

We have added this thoughtful comment (with slight changes) in the revised version to the Summary and Future Prospects section at the end of the manuscript.

6. Methods, page 16, what was the concentration of the RNA in the annealing solution? What was its concentration in the solution that was mixed 1:1 with reservoir for crystallization? What were typical dimensions of the crystals? Their morphology?

We have added the crystallization information in the Methods section of the revised version.

"The concentration of the RNA we prepared in the annealing solution is 0.4 mM. Then, we mixed 0.2 μ L of the RNA sample (the concentration is 0.4 mM) with an equimolar volume ratio of the reservoir solution. The typical dimensions of crystals are around 0.05-0.1 mm in a cuboid shape."

7. Regarding the "native" dataset, since there was quite a bit (150 mM) of Ca²⁺ present, did the authors inspect the anomalous difference Fourier for signs of Ca²⁺ binding to their imputed Mg²⁺ binding sites?

The coordination lengths between divalent metals and water oxygens is 2.4 Å for hydrated Ca²⁺ cations and 2.0 Å for hydrated Mg²⁺ cations. We assign the observed divalent cations to Mg²⁺ in the twister-sister ribozyme structure since we observe lengths of 2.0 Å. This statement has been added to the Methods section.

8. In the crystallographic methods, some indication of the quality of the experimental phases (for instance, phasing power, mean overall figure of merit prior to density modification) needs to be mentioned. Also, was density modification performed? With what program (most likely Resolve, if using PHENIX, but the authors need to say)? Also, what target function was used for refinement (specifically, was it a function with experimental phases)?

We used HKL2Map³⁶ to locate the positions of Ir(NH₃)₆³⁺ ions. The correlation coefficients output by SHELXD were 37.7 (all data) and 23.5 (weak data) (PATFOM of 10.3). Then we used programs SOLVE and RESOLVE in PHENIX Autosol³⁷ to solve the phases and do the electron density modification. Then, the model was initially built with the program PHENIX Autobuild³⁷ and manually built and adjusted using the program Coot³⁸. After that, it was used as the initial model to perform molecular replacement on the 2.0 Å native data set to solve the native twister ribozyme structure, followed by refinement of the structure with PHENIX³⁷ with a final R_{work}/R_{free} 0.22/0.24 (Supplementary Table 1). These changes have been added to the Methods section of the revised version.

9. Density modified (presumably) experimental electron density for the asymmetric unit needs to be shown in supplementary, overlaid on final refined model.

We added Supplementary Figure 7 in the revised version showing the 2F_o-F_c electron density map (contoured at 1σ) overlaid on our final refined model.

10. The mean precision of the refined crystallographic coordinates (Luzzati, sigmaA, etc) needs to be stated in methods.

The coordinates error of our refined model was estimated by maximum-likelihood method in PHENIX program. The value is 0.36 Å. We added this information to the Methods section of the revised version.

11. A side-by-side figure comparing 3D cartoon representations of the 3-way and 4-way TS structures should be included in supplementary

We have now done this in the revised version with stereo views showing comparisons of the 3-way and 4-way twister-sister ribozyme crystal structures in Supplementary Figure 5.

12. S Table 1, it is mean I over mean sigma of I, which is represented $\langle s(I) \rangle$ (with the s being a lower-case Greek sigma). Also, B-factors have units (angstroms squared). The refinement section should also have the highest-resolution shell values for resolution, number of reflections, and R values.

We are thankful to the reviewer for pointing out the mistake! We have corrected 's(I)' to 'σI', add B factors' units 'Å²' in the Supplementary Table 1. Besides, we added the resolution (2.07-2.0 Å), the number of reflections (2786), and R values (R_{work}/R_{free} 0.35/0.41) of the highest-resolution shell to the refinement result statistics.

13. The reference number 17 is incomplete.

Corrected in the revised version. Please note that the references have been renumbered in the revised version.

14. Figure 5b should be in a representation where the 2' substituents of the riboses are visible

Corrected in the revised version. Please see Figures 3c, d of the revised version.

Reviewer #3

Zheng et al. present the crystal structure of a twister sister ribozyme and present data to support the hypothesis that it utilizes metal ions in its catalytic mechanism. The crystal structure determination is of high quality and resolution. Zheng et al. compare this structure to an existing crystal structure of the twister sister ribozyme from David Lilley's group. The twister sister ribozyme structure from the Lilley group led to the hypothesis that this specific class of ribozymes are metalloenzymes. In contrast, the twister ribozyme was previously shown to employ nucleobases in the catalytic mechanism with no requirement for a metal ion. Zheng et al. aim to resolve this discrepancy by solving the structure of a different twister sister ribozyme. However, the conclusions derived from this analysis are relatively weak and uncertain. In my opinion, there was no clear biochemical or structural evidence for the involvement of a metal ion in the catalytic mechanism.

We have toned down our position of the role of divalent ions in twister-sister ribozyme catalysis in the revised version and instead highlighted the differences between the two structures of the twister-ribozyme.

We refer to our detailed responses to related issues raised by reviewer #1.

Major Points:

M1 is hypothesized to be the catalytic metal ion, however the binding sites for M1 in both twister-sister structures is not conserved. In the Lilley structure, it is forming a nucleobase interaction, whereas Zheng et al. observe inner shell coordinations to the phosphate oxygen atoms (of nucleotides G50 and C60). In addition, the location of the metal ion is different in both structures with respect to the cleavage site. Active site architecture is typically conserved between different enzymes employing the same mechanism. This active site architecture should be especially conserved between twister sister ribozymes since they are in the same class. This lack of conservation does not support the metalloenzyme hypothesis. In addition, the authors did not present definitive biochemical analyses to support this assertion. I would suggest phosphorothioate substitution and recovery to support the hypothesis that M1 is playing catalytic roles. I mention M1 since it is the only one in the active site forming inner shell coordination with G50 and C60, whereas M2 only interacts via outer shell coordination and is therefore difficult to probe. There was also no detailed mechanism illustrating the possible precise role of M1 in catalysis.

There is some confusion because what is labeled M1 in the David Lilley's structure is labeled M2 in our structure of the twister-sister ribozyme. Both structures show the water of a hydrated Mg^{2+} (labeled M2 in our structure and M1 in the Lilley's structure) to be within hydrogen bonding distance of the modeled 2'-OH of C at the C-A cleavage step.

To shed light on a possible role of the M1 ion, we synthesized RNA with a methylphosphonate backbone unit between C60 and U61 (C60dCmP) that is expected to interfere with the inner-sphere coordinations of the Mg^{2+} ions M1 and M4 that we see in the crystal structure. Despite this interference, we observed that cleavage was only slightly impaired in comparison to the wild-type activity. We must point out that we have used the

diastereomeric mixture of methylphosphonate RNA that accounts for maximal 50% disruption of the M1 and M4 coordination, respectively. The observed activity therefore might arise from the remaining occupied M1 and M4 sites. Clearly, thorough experiments that utilize diastereomerically pure methylphosphonate RNAs are needed in the future to give more precise answers on the putative role of M1 in catalysis.

Accordingly, we have toned down claims related to a direct participation of M1 in catalysis in the revised manuscript.

Minor Points

-incorrect use of the word “step” throughout the paper

It is not clear what “step” should be replaced by in the revised version. We had used the word “site” in our twister paper but were forced by a reviewer to change it to “step”.

We use “cleavage step” (= the two nucleosides linked by the scissile phosphate) to distinguish from “cleavage site” (= “cleavage step” AND the nucleosides and cations close by that interact with the cleavage step)

-the word “likely” is used too often in this paper

We used the word “likely” given some of the uncertainties in reaching definitive conclusions for certain segments of our contribution.

We also refer to our responses to reviewer #1.

-many grammatical mistakes

We have made an effort to locate and correct grammatical mistakes in the revised version. Thanks for pointing them out!

REVIEWERS' COMMENTS:**Reviewer #2 (Remarks to the Author):**

Point #9 of Reviewer 2's comments was apparently misunderstood by the authors. What is being requested is a experimentally-phased (after density modification) Fourier map, not a model-phased map with $2F_o-F_c$ coefficients. The point of the supplementary figure is to show the quality of the experimental phase calculation, not the quality of the model phases. Except for that, all of these referee's concerns have been addressed.

REVIEWERS' COMMENTS:

Reviewer #2 (Remarks to the Author):

Point #9 of Reviewer 2's comments was apparently misunderstood by the authors. What is being requested is a experimentally-phased (after density modification) Fourier map, not a model-phased map with $2F_o-F_c$ coefficients. The point of the supplementary figure is to show the quality of the experimental phase calculation, not the quality of the model phases. Except for that, all of these referee's concerns have been addressed.

Thanks for pointing out our misunderstanding! We prepared a new supplementary figure showing the final model with the initial experimentally-phased (after density modification) Fourier map.